# A data-driven network decomposition of the temporal, spatial, and spectral dynamics underpinning visual-verbal working memory processes

Chiara Rossi [1,2✉], Diego Vidaurre[3,4], Lars Costers[1,5], Fahimeh Akbarian [1,2], Mark Woolrich [4], Guy Nagels[1,6,7] & Jeroen Van Schependom [1,2✉]

The brain dynamics underlying working memory (WM) unroll via transient frequency-specific large-scale brain networks. This multidimensionality (time, space, and frequency) challenges traditional analyses. Through an unsupervised technique, the time delay embedded-hidden Markov model (TDE-HMM), we pursue a functional network analysis of magnetoencephalographic data from 38 healthy subjects acquired during an n-back task. Here we show that this model inferred task-specific networks with unique temporal (activation), spectral (phase-coupling connections), and spatial (power spectral density distribution) profiles. A theta frontoparietal network exerts attentional control and encodes the stimulus, an alpha temporo-occipital network rehearses the verbal information, and a broadband frontoparietal network with a P300-like temporal profile leads the retrieval process and motor response. Therefore, this work provides a unified and integrated description of the multidimensional working memory dynamics that can be interpreted within the neuropsychological multi-component model of WM, improving the overall neurophysiological and neuropsychological comprehension of WM functioning.

[1] AIMS lab, Center for Neurosciences, Vrije Universiteit Brussel, Brussels, Belgium. [2] Department of Electronics and Informatics (ETRO), Vrije Universiteit Brussel, Brussels, Belgium. [3] Department of Clinical Medicine, Center of Functionally Integrative Neuroscience, Aarhus university, Aarhus, Denmark. [4] Department of Psychiatry, Oxford Centre for Human Brain Activity (OHBA), Wellcome Centre for Integrative Neuroimaging, University of Oxford, Oxford, UK. [5] icometrix, Leuven, Belgium. [6] Department of Neurology, Universitair Ziekenhuis Brussel, Brussels, Belgium. [7] St Edmund Hall, University of Oxford, Oxford, UK. ✉email: chiara.rossi@vub.be; jeroen.van.schependom@vub.be

Working memory (WM) is a higher-order cognitive function that enables the temporary maintenance and manipulation of a limited number of items[1]. Any cognitive task, from language comprehension to mathematical reasoning, relies on WM, prompting neuroimaging research and neuropsychological understanding[2–4]. WM functioning is accomplished by the unfolding of cognitive subprocesses, i.e., stimulus encoding, maintenance, and retrieval, which unroll via information communication and integration. In neurophysiology, this translates into the temporary synchronization of long-distance oscillatory neural activities in specific frequencies, forming network connections[5,6].

These large-scale brain networks have first been uncovered by functional magnetic resonance imaging (fMRI) studies. In particular, the distinctive fMRI WM networks are the frontoparietal and the default mode network (DMN)[7–9]. Despite the great spatial resolution, fMRI only indirectly captures slow (a few seconds) fluctuations of neuronal activity. Instead, electro- and magnetoencephalography (EEG and MEG) directly detect neural activity with milliseconds (ms) temporal resolution—the timescale of cognitive processing steps[10].

Traditional electrophysiological studies have first disregarded the network dimension and investigated the temporal and spectral activation of specific brain regions. Event-related (ER) analyses of EEG time courses have identified task-locked temporal events such as the P300 (central-parietal regions), which has been used as a cognitive marker of WM[11–13]. Then, time-frequency analyses explored the spectral content underlying the ER waves, hence, the brain rhythms involved in WM processing. In particular, the prefrontal theta (4–8 Hz) was found to direct stimulus encoding, while the inhibitory occipital alpha (8–10 Hz) gates the incoming stimuli[14–16]. Recent M/EEG studies have started exploring long-distance frequency-specific or cross-frequency neural synchronizations underpinning WM[17]. For example, theta-gamma phase-phase coupling was associated with input-template matching[18,19]. On the other hand, frequency-specific (theta, alpha, etc.) phase-coupling was shown to differentiate different WM load conditions[20].

Aside from the neurophysiological description of WM, neuropsychology has conceptualized WM as a dynamic multi-component system, in which the communicating compartments —a central executive (attention control and encoding) and two slave storage units, the phonological loop and the visuospatial sketchpad[1,4,21]—operate the different WM processes[1,22]. Whereas neuropsychology provides an integrated picture of WM, neuroimaging research has explored the spatial, temporal, and spectral dimensions in a rather scattered way. Neurophysiological findings that focus only on one or two of these dimensions offer an incomplete view of WM and findings of different studies are difficult to merge.

The time delay embedded-hidden Markov model (TDE-HMMs) represents a potential alternative to investigate the WM network dynamics at 360°. This technique describes the experimental data as resulting from the alternating activation of hidden states[23]. One can understand the parallelism with neurophysiology: the recorded brain activity results from the recurring activation of brain networks (states) that we cannot directly observe. This method infers in an unsupervised manner a predefined number of states that depict power covariations and phase-coupling across regions throughout the data[24–26]. Therefore, the HMM states constitute spectrally defined functional networks that wax and wane over a timescale dictated by the experimental data. In fMRI data, the HMM states resembled the canonical resting-state networks[25]. In MEG data, although considering the inherent ambiguity of source-reconstructed data, these states can track the evolution of cognitive processes with great temporal resolution[26,27].

In our work, we apply the TDE-HMM aiming to fill the gap in the neuroimaging literature and describe the WM dynamics overarching the spatial (network), spectral, and temporal dimensions. We analyzed MEG data acquired during a WM paradigm, the visual-verbal n-back task. In one epoch of 1.4 s, we identify data-driven task-relevant states that portray how WM processes unfold. A theta prefrontal state performs early high-cognitive processing, an alpha temporal-occipital state rehearses the memory items, and a broadband frontoparietal state with an M300 temporal profile leads the manipulation and retrieval processes. These findings are consistent with the theoretical accounts of WM, unveiling traits of the WM dynamics, such as the M300 state. Altogether, our study provides a unified description of the time, space, and frequency profiles of the fast transient networks underpinning working memory.

## Results

**Participants**. We included 38 healthy subjects in this study (Table 1). The male (15 subjects) and female (23 subjects) groups do not significantly differ in age (one-way ANOVA test, $F = 4.3$, $p$-value = 0.05) or education (one-way ANOVA test, $F = 0.05$, $p$-value = 0.8). We report the mean reaction times (RTs) and the accuracy of response per paradigm condition in the Supplementary Figs. S1 and S2.

**Number of states, model reliability, and replicability**. We identified 6 as the optimal number of states as this setting could pick up the expected spatio-spectral traits of the WM task and minimize the redundant information across states[28]. We report the results of the 12 states inference in the supplementary materials section 2 (Fig. S3), to demonstrate the replicability of the relevant states identified in the 6 states inference, and to show the issues encountered with an increasing number of states.

To test the model reliability, we visually assessed the results of 4 different inferences (with 6 states), as suggested by ref. [29], and concluded that the model could consistently infer states with similar spatio-spectral traits. Additionally, we computed the temporal characteristics (lifetime, LT, interval time, IT, and fractional occupancy, FO) of the states over the concatenated data to verify that these results are consistent with what was previously reported by different HMM studies. We report this analysis in the supplementary materials section 3 (Figs. S4 and S5). The states' average lifetime is 73 ms, the average fractional occupancy is 18%, and the average interval time is 500 ms. These results are consistent with the HMM literature on resting-state and task data[26,28,29].

**Temporal dimensions—ER analysis**. Figure 1a shows the average activation across all paradigm conditions of the 6 HMM states during the n-back task. From this task-evoked activation plot, we identify the task-relevant states as those that are significantly activated or deactivated: states 1, 2, 3, and 5 (permutation test

**Table 1 Demographics of the dataset.**

|  | Number of Subjects | Age—years (mean ± std) | Education—years (mean ± std) |
|---|---|---|---|
| Male | 15 | 49.4 ± 6.9 | 14.9 ± 3.1 |
| Female | 23 | 42.7 ± 11 | 14.7 ± 3.3 |
| *p*-value |  | 0.05 | 0.8 |

For each group, the age and education are expressed as mean and standard deviation (std). The *p*-values result from a one-way ANOVA test for age and education, considering sex as a grouping variable, with levels 'Male' and 'Female'.

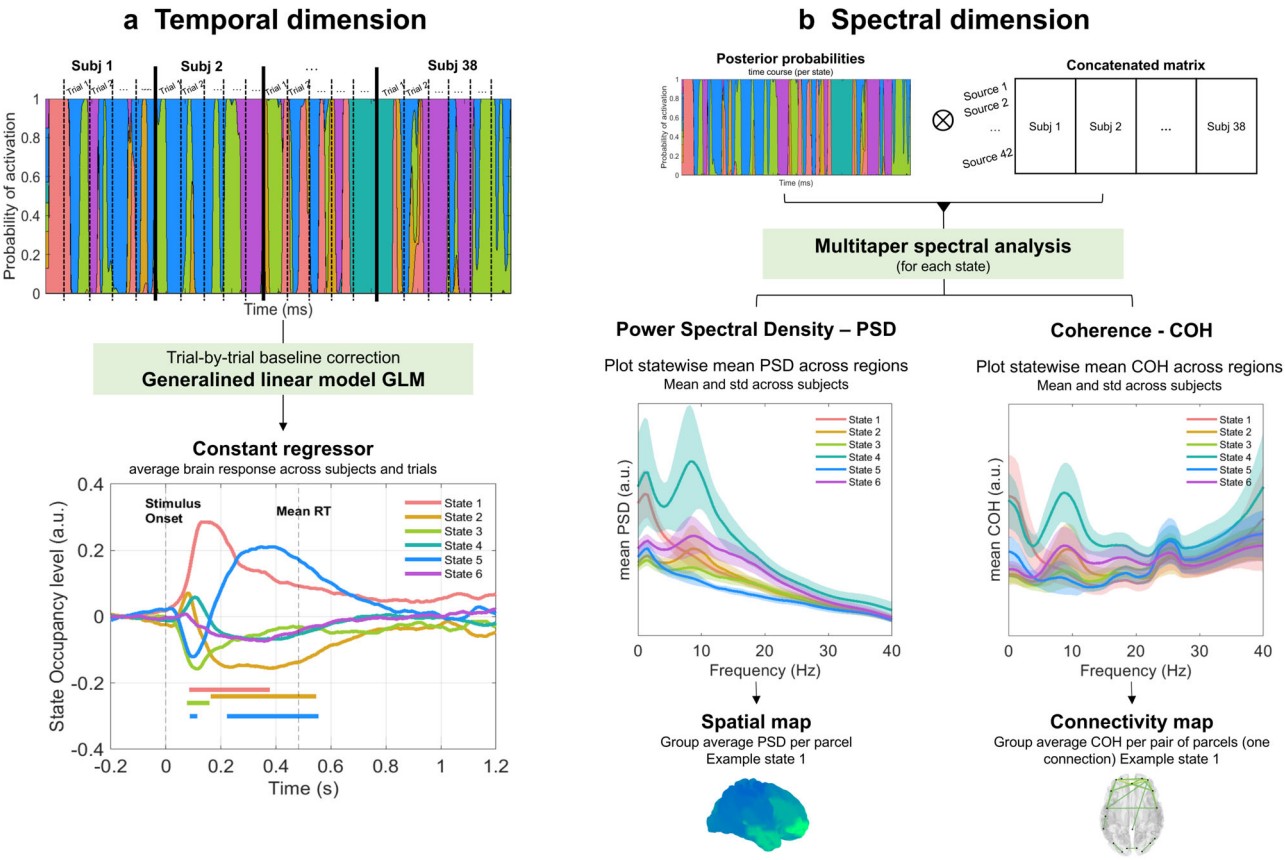

**Fig. 1 Methodological steps to extract the states' temporal, spectral, and spatial profiles. a** Temporal dimension. Starting from the states' time courses—the posterior probabilities—we epoched the time series with respect to the task information (stimulus onset) to define the trials using a [−0.2, 1.2] s window. We ran the generalized linear model (GLM) to statistically evaluate the increased or decreased modulation of the statewise fractional occupancy (activation) level compared to the baseline. We report the states' average response across conditions (provided by the constant regressor). The straight lines in the plot indicate the time points where the state of the same color is significantly activated or deactivated (permutation test, number of permutations = 1000, $p < 0.025$, correction for multiple comparisons via maximum statistics). The significantly modulated task-relevant states are states 1, 2, 3, and 5. **b** Spectral dimension. We weighted the MEG recordings by the states' time courses and used a multitaper to compute the spectral density of the weighted MEG data for each subject and state separately. From this, we extracted the power spectral density (PSD) over the brain, which constitutes the spatial map of activation of a state, and the coherence across regions which constitutes the phase-coupling network of a state. We reported the plot of the statewise PSD averaged across regions over the broad frequency spectrum (1–40 Hz)—the bold lines display the mean across subjects, and the lighter area includes the standard deviation across subjects. The same plot is reproduced for the phase-coupling averaged over all the connections in the broadband spectrum (1–40 Hz)—the bold lines show the mean coherence across subjects, and the area represents the standard deviation.

with number of permutations = 1000, $p < 0.025$, multiple comparison correction by maximum statistics). The increased or decreased activity is observed individually for each state, and it describes the modulation of the statewise occupancy level after the stimulus onset compared to the baseline level. When the task-evoked activity of a state with a strong phase-synchrony goes below baseline, other states (with weaker phase-synchrony in the same frequency band) are activated more frequently. Therefore, the average estimated phase-coupling in the determined frequency range at that timepoint is below average and, hence, suppressed. The full description of all 6 states is reported in the supplementary materials (Figs. S6–S11); the non-task-relevant states are associated with resting-state and baseline activity[29]. Figures 2 and 3 report the characteristic profiles for the task-relevant states.

**Spectral modes.** Figure S12 in the supplementary materials reports the four spectral modes in which the states' spectral content is factorized via non-negative matrix factorization (NNMF): spectral mode 1 is associated with the activity in the low frequencies (1–8 Hz), spectral mode 2 with the alpha (8–12 Hz)

band, and spectral mode 3 with the beta (12–25 Hz) band. Spectral mode 4 includes activity within the low-gamma spectrum (25–45 Hz). The model we implemented, the TDE-HMM, is biased towards low frequencies because of the use of PCA, which discards higher frequencies[23]. However, intermediate, non-discarded frequencies might be overemphasized with respect to the lowest frequencies[32]. We decided to disregard this spectral mode in our interpretation because of this consideration and our focus on large-scale brain networks. In fact, long-distance synchronizing neuronal activities have been previously associated with lower rhythms (theta and alpha)[6,30].

**States description.** We present each task-relevant state by means of (1) its time course of activation (state task-evoked response or occupancy level) for all the paradigm conditions, (2) the mean z-score power spectral density (PSD) distribution over the brain, and (3) the phase-coupling network over the brain.

*State 1—The theta prefrontal state.* State 1, depicted in Fig. 2 (panels a, b, c, and d), represents an early low-frequency frontal network. This state is significantly activated between 150 and

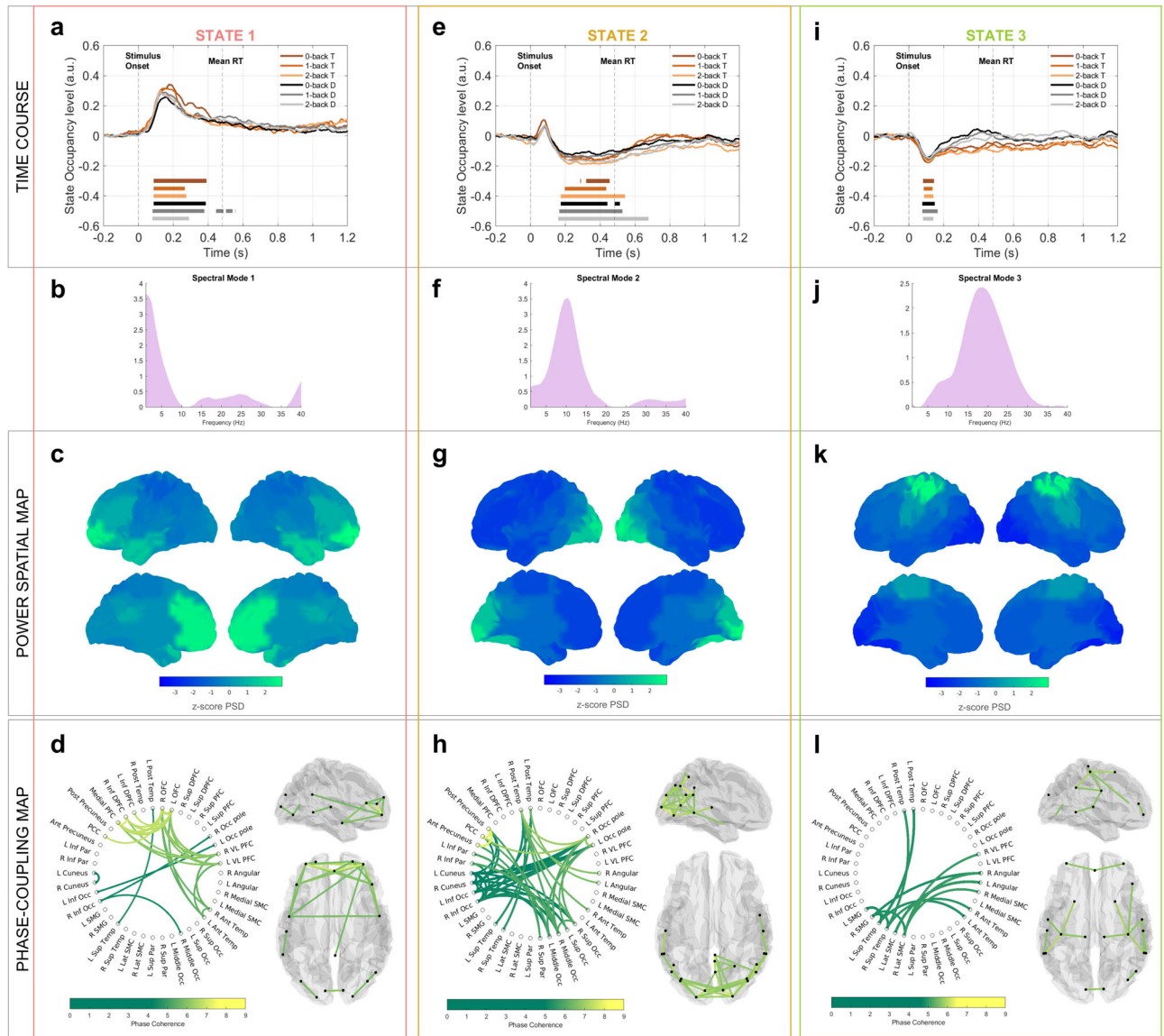

**Fig. 2 Temporal, spectral, and spatial description of the task-relevant states 1, 2, and 3.** Panels **a** (state 1), **e** (state 2), and **i** (state 3) report the states' time course (task-evoked response or occupancy level) resulting from the GLM analysis for the 6 paradigm conditions, separately. In each plot, the flat lines show the time points in which the statewise occupancy level for the specific task condition (color-coded) is significantly different from the baseline level (permutation test, number of permutations = 1000, *p* < 0.025, correction for multiple comparisons via maximum statistics). Panels **b**, **f**, and **j**, present the spectral mode associated with the states 1, 2, and 3, respectively. Boxes **c** (state 1), **g** (state 2), and **k** (state 3) show the *z*-score mean power spectral density (PSD) map, whilst boxes **d** (state 1), **h** (state 2), and **l** (state 3) present the phase-coupling networks (brain glass and circular graph) in which only the phase-coupling connections surviving thresholding are plotted. Only the phase-coupling connections surviving thresholding are plotted. Figure 5 reports the description of state 3.

350 ms (peak at 200 ms) peristimulus time (PST) in all paradigm conditions, Fig. 2a. It presents a low-frequency peak of both PSD and coherence (Fig. 1b). The low-frequency peak of mean PSD appears in the right and left orbitofrontal cortices (OFCs), the medial prefrontal cortex, the right and left anterior temporal cortices, and the posterior cingulate cortex (PCC), Fig. 2c. The connectivity network shows strong phase-synchronization in the low frequencies between prefrontal and anterior temporal regions and a connection between the OFC and the PCC, Fig. 2d.

*State 5—The M300 state.* State 5, reported in Fig. 3, displays a broadband and spatially complex network. The task-evoked response presents a negative peak around 100 ms PST that is significant only for the 2 back conditions (target and distractor), Fig. 3a. Afterward, the activation level steadily increases, and the

state is significantly activated between 225 and 500 ms PST. The state presents high mean phase-coupling around 25 Hz (Fig. 1b) referred to as spectral mode 3, instead, the mean PSD does not present any frequency-specific peak (Fig. 1b). Therefore, we refer to this state as a broadband state. Considering the spectral content of spectral mode 3 (Fig. 3d), the phase-coupling network displays connections between the anterior and posterior precuneus and the PCC, and the right and left OFC with the medial PFC. In the same spectral mode, the mean PSD covers a wide area of the frontal cortex, including the inferior and superior dorsal PFC, the medial PFC, the left and right superior PFC, and the right and left medial sensorimotor cortex (SMC).

*State 2—The occipital state.* State 2, displayed in Fig. 2 (panels e, f, g, and h), represents an occipital alpha network. After a

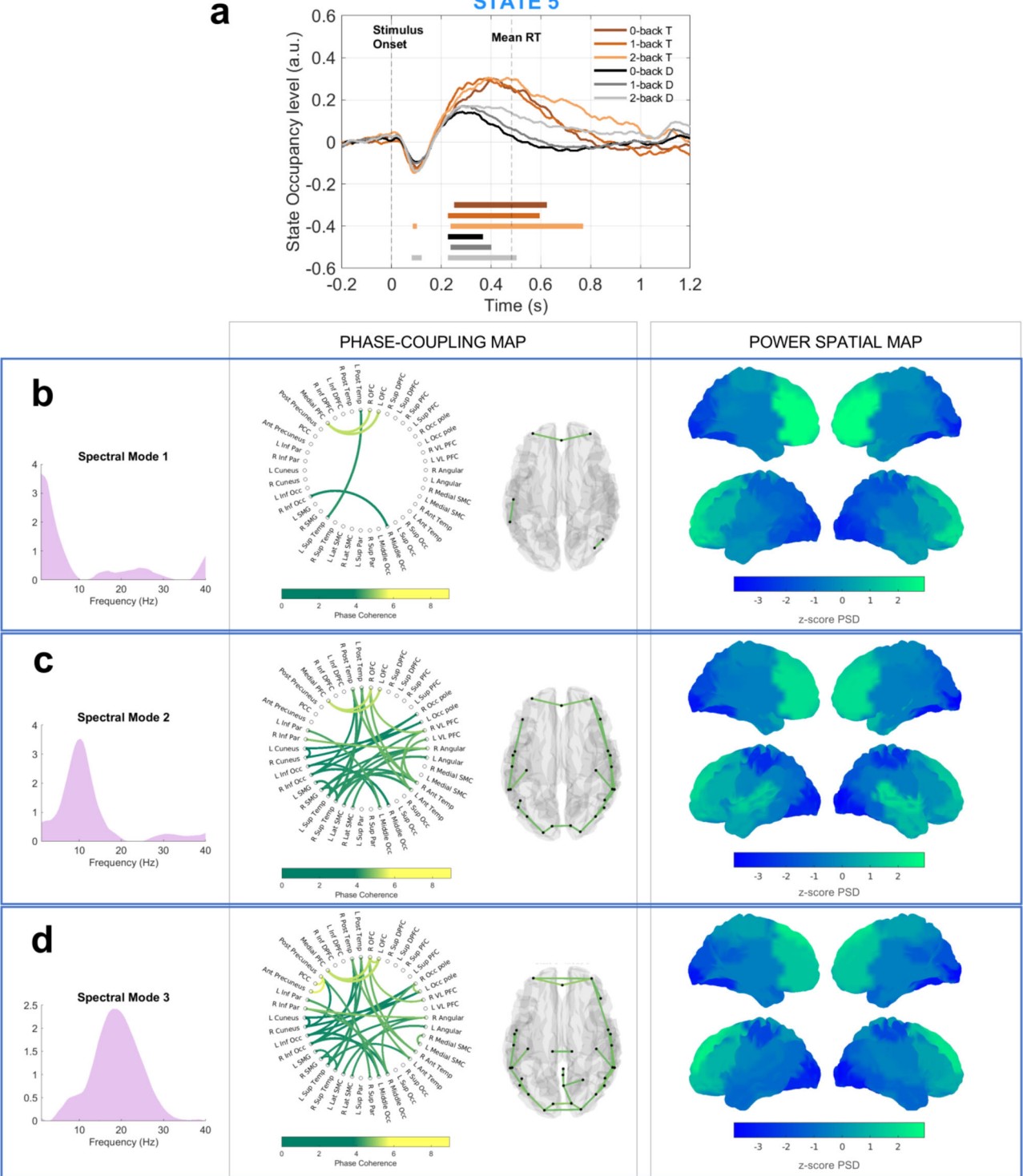

**Fig. 3 Temporal, spectral, and spatial description of the task-relevant state 5. a** The plot shows the task-occupancy level of the state, resulting from the GLM analysis; we plot all the six paradigm conditions, separately. The flat lines show the time points in which the state occupancy level for the specific task condition (color-coded) is significantly different from the baseline level (permutation test, number of permutations = 1000, p < 0.025, correction for multiple comparisons via maximum statistics). Panel **b** reports the spectral content related to spectral mode 1, panel **c** reports the spectral content referred to spectral mode 2, and panel **d** shows the spectral content for spectral mode 3. Each panel includes the profile of the spectral mode showing which frequency range is considered, the phase-coupling network (brain glass and circular graph) in which only the phase-coupling connections surviving thresholding are plotted, and the z-score mean power spectral density (PSD) map. The other states are reported in Fig. 2.

non-significant peak of activation around 100 ms, the state task-evoked response significantly decreases between 200 and 500 ms after stimulus onset for all task conditions, Fig. 2e. The activity of this state arises primarily in spectral mode 2, as it shows a peak of

mean PSD and coherence around 10 Hz (Fig. 1b). The mean PSD distribution spatially focuses on the occipital lobe, and the left and right precuneus, Fig. 2g. The phase-coupling network reveals the synchronization in the alpha band of occipital regions with

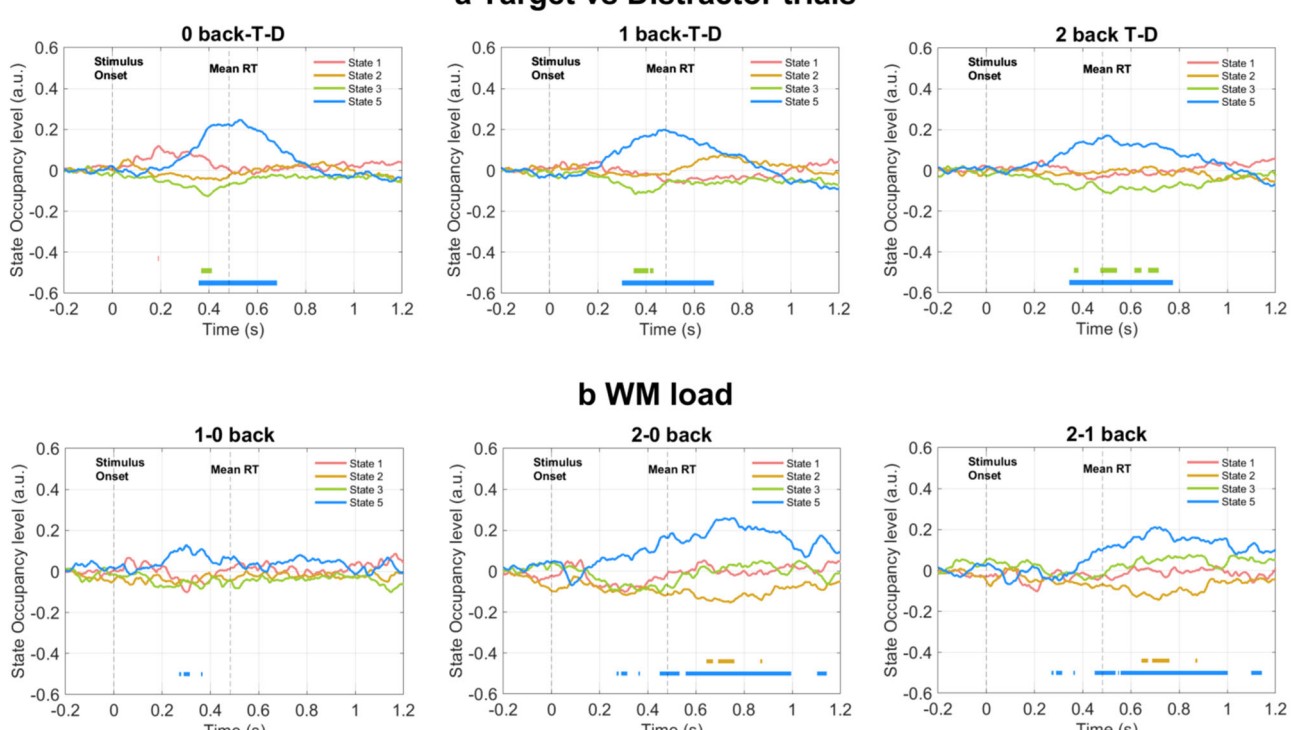

**Fig. 4 States' task-evoked activity modulated across paradigm conditions. a** Target versus distractors. The graphs report the contrast regressors to evaluate the state activation difference between target (T) and distractor (D) trials. Each plot considers a single load condition: 0, 1, and 2. **b** WM load. Panel **b** shows the contrast regressors evaluating the difference between all pairs of WM load conditions: 0, 1, and 2. The straight lines in each plot denote the time points when the state's contrast regressor (of the same color) is significantly (permutation test, number of permutations = 1000, $p < 0.025$, correction for multiple comparisons via maximum statistics) modulated compared to the baseline.

some parietal ones, such as the PCC, the angular gyri, and posterior temporal regions, Fig. 2h.

*State 3—The sensorimotor state.* State 3, shown in Fig. 2 (panels i, j, k, l), represents a sensorimotor network. Its spectral content lays mostly around 25 Hz (Fig. 1b), therefore, in spectral mode 3. It displays a significant peak of deactivation at 100 ms peristimulus, Fig. 2i. The PSD map reveals high beta activity in the left and right supramarginal and the left and right sensorimotor cortices, Fig. 2k. The phase-coupling network shows broad connections that branch further than the regions we identified in the PSD map. We observe intra- and inter-hemispheric connections between the left and right posterior temporal regions, the left and right angular gyri, the left and right supramarginal, and the left and right sensorimotor cortices, Fig. 2l.

**Task-related modulation.** Figure 4a shows the difference between the states' task-evoked response during target and distractor trials. State 1, the prefrontal theta state, presents a small but significantly higher peak of activation at 200 ms in the 0 back target compared to the 0 back distractor condition. State 3, the sensorimotor state, shows a significantly decreased task-evoked response in the target compared to distractor trials around 400 ms in the 0 and 1 back conditions, and between 400 and 700 ms in the 2 back conditions. Last, state 5, the M300 state, presents a significantly amplified (+20 %) task-evoked response in target than distractor trials between 300 ms and 700 ms in all WM load conditions.

Next, we investigated the difference between the states' task-evoked response between WM load conditions (1–0, 2–0, and 2–1), Fig. 4b. The analysis incorporated all the target and

distractor trials for each WM load condition. The activation of state 5 is significantly increased in the 1 back condition compared to the 0 back condition around 300 ms, and also in the 2 back condition compared to both 1 and 0 back conditions between 400 and 1000 ms after stimulus onset. Instead, the occupancy level of state 2 is significantly reduced in the 2 back conditions compared to 0 and 1 back conditions between 650 ms and 750 ms after stimulus onset.

## Discussion

Working memory (WM), a high-order cognitive function, comprises different subprocesses (i.e., encoding, maintenance, retrieval) that are executed by transiently synchronizing neural populations, forming dynamic functional networks[6,22]. This work investigates magnetoencephalographic (MEG) data acquired during a visual-verbal n-back task and utilizes the time delay embedded-hidden Markov model to extract, in a fully data-driven way, transient and spectrally resolved networks. As a result, we obtain an integrated description of the temporal, spectral, and spatial dimensions of the waxing and waning networks underpinning WM processing, as summarized in Fig. 5.

The n-back task has repeatedly proved to elicit a robust neural activity consistent within and between MEG recording sessions[31–33]. The task design is such that all the WM processes unroll in a single time window. Therefore, we cannot univocally associate one state with a single WM process. However, we interpret each state starting from its spatio-spectral traits and their functions in the WM literature.

State 1 depicts the early (ca 180 ms after stimulus onset) rising of low-frequency activity in the prefrontal, anterior temporal, and posterior cingulate (PCC) regions. We associate the

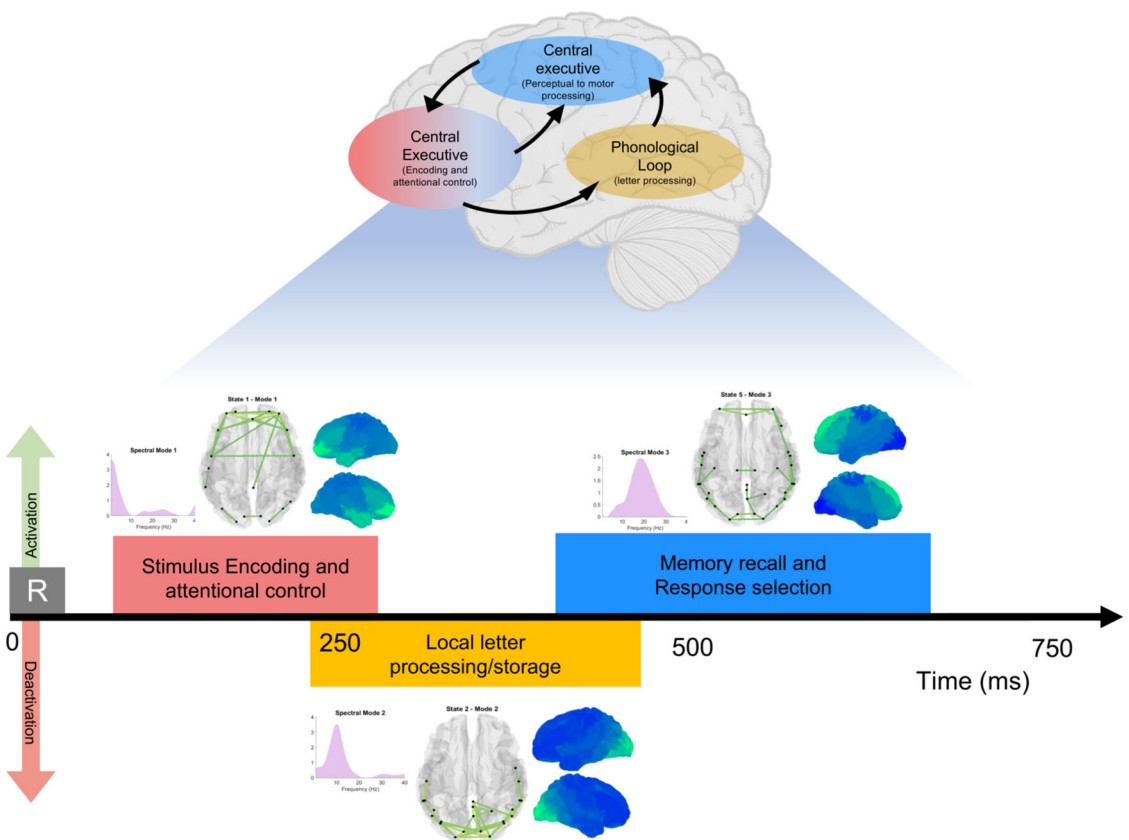

**Fig. 5 Overview of the main results of this study.** We report a schematic representation of the WM multi-component model as presented by refs. [4,21], and then we depict how this work decomposes the working memory network dynamics and provides a multidimensional description of WM during an n-back task. The color of each unit is associated with the color of the state that plays the same role. We do not aim at creating a one-to-one correspondence between the data-driven results and the neuropsychological model but rather present an integrated understanding of the WM dynamics. The HMM states provide a spatial, spectral, and temporal representation of WM dynamics, improving the overall description of WM.

low-frequency mode with the theta (4–8 Hz) rhythm, consistently reported in the WM literature[34–36]. Mainly detected in the prefrontal cortex and hippocampus, WM theta activity has been related to integrating and controlling functions during higher-order cognitive processing[37–39]. State 1 exhibits this prefrontal theta activity, suggesting a role as an early high-order stimulus processor and encoder.

Interestingly, state 1 displays theta synchronization between the orbitofrontal cortex, OFC, and PCC. This has previously been interpreted as a volitional top-down attentional control mechanism[40]. This OFC-PCC connection is also detected in the anterior default mode network (DMN) extracted in resting-state data[25]. fMRI WM studies have extensively explored the role of the DMN in WM, reporting this network as actively engaged during WM encoding[7,8]. Furthermore, the OFC is also theta-coupled with anterior temporal regions, identified as the maintenance regions for verbal stimuli[41]. This coupling could then embed the attentional control of the OFC onto the temporal cortex during WM maintenance, as previously reported by ref. [42].

Therefore, we hypothesize that state 1 extracts the mental representation of the stimulus during encoding and exerts top-down attentional control onto low-level stimulus processing (PCC) and maintenance (temporal cortex). In Baddeley's neuropsychological model, these functions (among others such as inhibition, manipulation, and shifting) are assigned to the executive control unit[1,22], in which Cowan identifies attention as the core component[3].

We observe that the task-evoked response of state 1 does not significantly modulate with WM load. Instead, a small modulation appears in the contrast between target and distractor trials in the 0 back (Fig. 4a). This paradigm condition resembles the oddball task, designed to investigate attention. We hypothesize that state 1 might be more driven by attentional requirements, which do not change significantly between different conditions in the n-back paradigm, rather than working memory processing demand, that, instead, we associate with state 5 (as discussed below).

The network configuration reveals connections between regions with different roles (for example, OFC attention and anterior temporal maintenance) that could depict the interplay between different processes. This reflects the overlapping unrolling of WM processes during the n-back task and might represent the dynamic communication between the executive unit and the slave components in the neuropsychological model.

State 5 exhibits a broad spectral activity distributed in a complex network involving frontal, temporal, and parietal regions. Like state 1, state 5 also recruits frontal regions, in particular the dorsolateral PFC, with rising theta activity, which are generally associated with high-order cognitive functions and largely recruited in WM[43]. Scharinger et al. suggested that the n-back task requires high executive processing demand compared to other WM tasks[12], which could explain the sustained recruitment of the prefrontal cortices throughout the epoch in states 1 and 5. However, the functions carried out by the two states differ depending on the timing of activation, the frequency-specific connections formed between the prefrontal and other regions, and the statewise modulation throughout the task.

The evoked response of state 5 resembles the P300 wave, a task-locked temporal event that characterizes the n-back response

and is detected in central-parietal regions (in EEG data)[12,44]. This feature has been consistently associated with the retrieval process, in which perceptual (recalling, matching, updating) and motor (response selection) processing are carried out[45]. While the P300 has been extensively studied in the EEG literature[12,46], its magnetic counterpart, the M300, has been only recently assessed in MEG data. Via a single region event-related analysis, Costers et al. have identified the temporal cortex and the SMC—regions also recruited in state 5—as the source of the M300 activity[47].

The task-locked occupancy level of state 5, the M300, is modulated by WM load (Fig. 4b). The increasing task difficulty requires increasing resources in the matching and recalling processes that we observe resulting in an increased M300 amplitude. Jensen et al. have reported an increased theta activity with increasing WM load during the retention stage[35], not in the early encoding step, supporting the increased M300 amplitude and justifying the different behavior we observe between states 1 and 5. While the M300 wave shows an increased amplitude, the P300 amplitude decreases with increasing WM load[12,44,48], and this opposite effect might be caused by the physical nature of the signal, or the different perspective of analysis (source-reconstructed MEG functional networks versus sensor-level single region EEG data).

The P300 wave has been associated with a wide broadband 25–40 Hz activity[11,49], and the same alpha/beta activity rises in state 5, in which a frontal-temporal coupling links the executive regions (PFC) and memory storage (temporal regions, as discussed in state 2). Quentin et al. suggested that, during WM retrieval, stimulus-template matching would be carried out by executive prefrontal regions rather than the maintaining temporal regions, to preserve the memory content[50–53]. The same connection might also be involved in updating the memory template. The activation of state 5 is significantly amplified and prolonged in the target as compared to distractor trials (Fig. 4a), and this difference might reflect the passage from perceptual to motor processing after target recognition. In fact, we observe that parietal regions (such as the medial SMC and the supramarginal regions) displaying beta activity are also recruited in state 5, and they have previously been detected in response selection and motor planning[54].

In conclusion, we theorize that the executive unit of WM could be depicted by two states: state 1 for early stimulus encoding and attentional control, and state 5 for manipulation and response processes.

State 2 is characterized by an alpha-dominant occipitoparietal network with phase synchronization between the posterior and superior temporal and occipital regions. The state's evoked-response is significantly decreased between 200 and 500 ms PST (Fig. 2e), resembling the event-related desynchronisation (ERD) of the occipital alpha activity that several WM neuroimaging studies detected during and following the early stimulus encoding phase[12,15,20,55]. The alpha activity in the temporal and occipital fusiform regions was observed to decrease with increasing local letter processing and word awareness[47,56]. The same regions were also reported to conduct letter processing and maintenance, specifically in the n-back task with visual-verbal stimuli[57], and Lochy et al. observed the left ventral occipital-temporal cortex involved in letter representations[58]. Therefore, the suppressed alpha in state 2 could reflect local independent letter processing, as traditionally reported in the WM literature. The evoked-response of state 2 is modulated by WM load, and with increasing WM load—increasing processing demand—state 2 is suppressed for a longer time. The same effect is consistently observed for the alpha occipital ERD wave, and this observation corroborates the interpretations of state 2 as local letter processing.

In the visual-verbal n-back task, the verbal nature of the stimulus (the letters) determines how the information is stored and maintained[1,59,60]. As suggested in Baddeley's neuropsychological model, the visually acquired letter is translated into its phonological representation and is stored as such in the phonological loop[1]. This compartment has been linked to language processing areas (e.g., Broca's area and temporal cortex), which are recruited in state 2[4]. Therefore, in state 2, the alpha phase-coupling between occipital and temporal regions could represent the first visual (occipital) processing being translated in—and then stored as—the correspondent phonological representation (temporal).

With all the attentional resources allocated to encode the stimulus, other activities should be suppressed[12]. In this regard, we observe that the fractional occupancy level of the beta sensorimotor state, state 3, is significantly suppressed (around 100 ms post-stimulus presentation) compared to the pre-stimulus baseline. The model inferred the sensorimotor state consistently with previous TDE-HMM analyses[24,29]. The spectral profile arising in the beta band is interpreted as the rhythms generated in the SMC[61]. The suppressed beta activity in state 3 prevents the ongoing encoding of WM representation from disruption[54,62]. The prolonged suppression of this beta rhythm in target than distractor trials (see Fig. 4a) could reflect the increased attentional and processing demand leading to the motor response in target trials[63].

This work presents a few limitations that call for improvement in future works. The TDE-HMM assumes that all the inferred states are activated in a mutually exclusive fashion. However, the brain likely recruits different brain networks simultaneously. More recent analysis designs, like DYNEMO[64], could overcome this limitation. Regarding the spectral dimension, we address two aspects. First, gamma band has been consistently reported to play an important role in WM maintenance. Because of methodological limitations, we discarded this frequency range. However, future work may investigate the occurrence and role of gamma in the WM network dynamics. Secondly, in the introduction, we mentioned the late developing research line on cross-frequency coupling. Future studies should investigate the cross-frequency coupling within each state, in particular, for state 5, in which the broadband activity might hide cross-frequency coupling. In addition to the n-back, the WM literature presents other paradigms to address working memory. Each task could yield slightly different dynamics, as they recruit WM subprocesses differently[38,62]. To validate this experimental design, future works should apply the same methodology to investigate other working memory tasks (the Sternberg task or the symbol-to-digit modality test); this could help identify the task-specific from the general WM processes.

To conclude, our work explored the n-back WM dynamic in MEG data utilizing the TDE-HMM technique to extract data-driven dynamic functional networks[29]. The model inferred four task-relevant states with unique temporal, spatial, and spectral profiles that provide a unified and integrated description of the multidimensional nature of the WM network dynamics. We are able to interpret the HMM states within Baddeley's multi-component model of WM[1]. This unique exploration of WM reveals traits such as the M300 state that represents a potential magnetic counterpart of the cognitive EEG P300 feature and could lead to a more in-depth understanding of cognitive processes in MEG data.

## Methods

**Participants**. The dataset includes 38 healthy subjects with normal to corrected vision. All participants signed an informed consent, and the study was approved by the ethics committees of the National MS Center—Melsbroek and the University Hospital Brussels (Commissie Medische Ethiek UZ Brussel, B.U.N. 143201423263, 2015/11). All ethical regulations relevant to human research participants were followed.

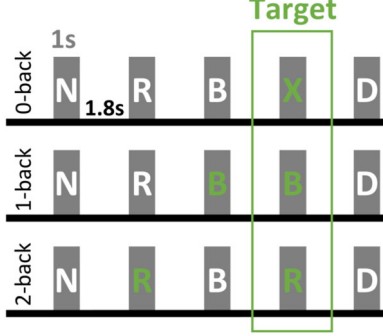

**Fig. 6 Graphic representation of the visual-verbal n-back task.** Every letter is displayed for 1 s, and the inter-trial period is 1.8 s. Every green letter represents the template (target) to which every stimulus is matched, and the target letters—for which subjects must press a button—are in the green rectangle. The white letters are, instead, considered distractors.

The demographics of the population are reported in Table 1. The statistical analysis of the demographics (age and education) was conducted via a one-way ANOVA test, considering sex as the grouping variable.

**Data acquisition**

*Magnetoencephalographic (MEG) data.* Every subject underwent an MEG and MRI acquisition. The MEG data were acquired at the CUB Hôpital Erasme (Brussels, Belgium) with two scanners: the Neuromag VectorViewTM system (13 subjects), and then the updated NeuromagTM TRIUX system (MEGIN Oy, Croton Healthcare, Helsinki, Finland) (25 subjects). Both MEG set-ups consist of 102 triplets of sensors, each including 2 planar-gradiometers and 1 magnetometer. The device lays in a light-weight magnetically shielded room (MSR, MaxshieldTM, MEGIN Oy, Croton Healthcare, Helsinki, Finland). Prior to the MEG recording, the shape of the subject's scalp was recorded using an electromagnetic tracker (Fastrak, Polhemus, Colchester, Vermont): several points were traced over the whole scalp, nose, and face, in addition to the 3 fiducial points (nasion, left and right preauricular). During the MEG acquisition, participants sat in the MEG scan with 3 coils on the mastoid, left, and right forehead to track the head's movements; additional sensors recorded the electrocardiogram (ECG) and electrooculogram (EOG). The MEG signal was acquired with a sampling frequency of 1000 Hz, and a [0.1 330] Hz band-pass filter.

*Magnetic resonance imaging (MRI) data.* The MRI data were collected at the Universitair Ziekenhuis Brussel (Jette, Belgium), using a 3 T Achieva scanner (Philips, Best, Netherlands). The 3D MR images were T1-weighted (longitudinal MRI with the subject in Head First-Supine, HFS, position). The scan used an echo pulse sequence gradient with Echo sequence TE 2.3 s, the recording parameters were TR (repetition time) = 4.939 ms, flipping angle 8, $230 \times 230$ mm$^2$ field of view, 310 sagittal slices, resulting in a 0.53 by 0.53 by 0.5 mm$^3$ resolution. This image was affinely coregistered to the MNI152 atlas. The structural and functional acquisitions were collected with 5 days (median value) in between (IQR 2–10 days). In this work, the MRI data were only used to perform an accurate source-reconstruction of the MEG data by co-registering the MEG data to the subject-specific MRI.

**Task design.** All participants performed a visual-verbal n-back task during the MEG recording. This paradigm consists of showing a sequence of letters, and the subject is instructed to respond to a target letter by pressing a button with the right hand.

In the 0-back condition, the letter X is the target; during the 1 and 2-back conditions, the target is any letter that coincides with the $n^{th}$ ($n = 1,2$) preceding one. The rest of the shown letters are considered distractors. Figure 6 visually explains the task.

The experimental arrangement consisted of a screen 72 cm from the MEG helmet. Letters projected on the screen fit within a $6 \times 6.5$ cm$^2$ area. A photodiode detected the stimulus onset. The reaction time was then computed as the time between the stimulus onset detected by the photodiode and when the subject pressed the button.

Participants did a training session before the recording to verify whether they understood the task. Twelve blocks of 20 letters (stimuli) each were presented pseudo-randomly, four for each paradigm condition. The total number of target trials is 25, 23, and 28, for the 0, 1, and 2-back conditions, respectively.

**MEG data preprocessing**

*Data preprocessing.* The entire analysis was developed in MATLAB 2020b, and the data preprocessing was carried out following the MEG analysis pipeline proposed in ref. [29]. The pipeline uses Oxford's Software Library and builds upon SPM12 (Welcome Trust Center for Neuroimaging, University College London) and Fieldtrip[65]. First, we coregistered the MEG data to the T1 MR image of the same subject, applying the RHINO algorithm. Here, we used the subject-specific fiducial points acquired with the Polhemus tracker, to minimize the coregistration error. Next, we downsampled the MEG data to 250 Hz and applied a band-pass filter [1, 45] Hz—Butterworth IIR filter of order 5 with zero-phase forward and reverse filter, the instability is solved by reducing the order of the filter—to discard the high and low-frequency noise. We also included a notch filter at 50 Hz to remove the remaining power line effect, which could represent a source of noise for the HMM inference. We then performed artefact rejections. First, data segments of one second with an outlier standard deviation were discarded. Next, we applied the AFRICA algorithm that decomposes the data into 62 independent components (ICA) and removes those that correlate with ECG and/or EOG ($r > 0.5$). In the last step, we again visually examined the data to verify that all major artefacts were removed.

The MEG sensor data were normalized across sensor types (magnetometers and gradiometers) via eigenvalues decomposition[66]. Afterward, we applied the linearly constrained minimum variance (LCMV) beamforming algorithm to project the sensor data onto the source space[66]; the source reconstruction was based on a single-shell forward model in MNI space with a projection on a 5 mm dipole grid.

*Parcellation.* To parcel the source-reconstructed data, we used a 42 cortical regions atlas used before in refs. [67,68]. The time course for each region of interest (ROI) was extracted as the first principal component across the voxels' time series. As beamforming may lead to signal leakage between regions, we orthogonalized the parcels' time series by multivariate symmetric leakage correction[69]. This conservative approach discards zero-lag components between neighboring regions, assuring that the following connectivity analysis is not affected by volume conduction.

*Sign-flipping.* After beamforming, the sign of the dipoles is arbitrarily assigned, and this hinders the analysis across subjects and across brain regions[70]. Therefore, we applied the sign-flipping algorithm presented by Vidaurre et al.[68].

**Time delay embedded-hidden Markov model (TDE-HMM).** In this section, we explain the different aspects of the TDE-HMM model. For the mathematical formalism and a detailed

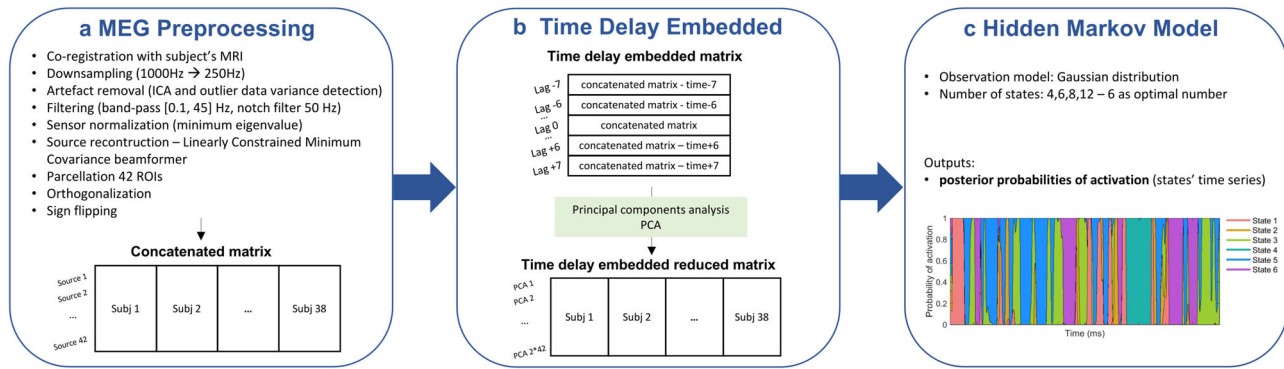

**Fig. 7 Methodological steps of the preprocessing pipeline and the TDE-HMM inference.** In box **a** we list the preprocessing steps followed to prepare the MEG data and prior to the model inference. Box **b** reports the main steps of the time delay embedding analysis to build the embedded matrix used as input to the HMM model. Box **c** presents the main aspects of the HMM inference and the results: the statewise posterior probabilities or state time course.

explanation of the different algorithms, we refer to refs. [23–25]. The application of this model follows the guideline paper[29].

*Hidden Markov model.* The hidden Markov models (HMMs) assume that the observed data result from the alternating activation of a discrete number of hidden states, which in this application represent large-scale brain functional networks. The Markovian constraint stipulates that the state activated at time t depends only on the observed data at time t and the hidden state at time t-1. The relationship between the hidden state and the observed data is ciphered in the observation model, a multivariate autoregressive (MAR), or a Gaussian distribution (such as in this application) of the observed data. Each state is defined by a specific set of distribution coefficients (mean activation and covariance matrix), computed via Bayesian inference[24,28]. Vidaurre et al. have implemented a stochastic algorithm that iteratively computes the states' parameters over a subset of subjects[68]. This approach compromises between an optimal inference and reduced computational time and load. The states' inference is run on a matrix in which the MEG preprocessed recordings of all the subjects are concatenated. Therefore, the states are inferred at the group level, enabling a direct comparison across subjects.

The model requires to set apriori the number of states to infer. As mentioned in Fig. 7c, we ran multiple inferences with 4, 6, 8, and 12 states to assess the model's behavior and the replicability of the results.

*Time delay embedded—HMM.* If we considered an observation MAR model with an autoregressive order $r > 0$, we would realistically capture the historical temporal interactions between time series of different brain regions. The latter is not instantaneous but delayed because of the finite conduction speed in the communication between neural populations. However, the huge number of parameters to compute (r*ROIs*ROIs) is computationally expensive and could lead to overfitting[25]. Vidaurre proposed to embed the conduction delay in the input matrix and consider as an observation model a Gaussian distribution.

Figure 7b shows how the embedded matrix is built starting from the concatenated original matrix. Similarly to ref. [25], we considered 15 lagged versions of the original data matrix with [−7: +7] lagged points, corresponding to a time window of 60 ms. These matrices are piled onto the original concatenated matrix, assembling an embedded matrix that is then reduced by applying principal component analysis (PCA) and extracting the 84 (2*ROIs) principal components. This reduced matrix is computationally lighter and includes, indirectly, information on the historical interactions between brain regions. The number of lags and principal components is determined to make the model

more sensitive to lower frequencies (theta and alpha, 8–16 Hz)[25], which are the core rhythms of WM.

*Posterior probabilities—states time courses.* Once the states are uncovered (via stochastic Bayesian inference), the following step consists of extracting the statewise posterior probabilities—the timepoint by timepoint probability that a certain state is activated given the related observed data. These constitute the states' time courses and are computed through the forward and backward algorithm[23].

**Reproducibility.** To test the reproducibility of the results and the reliability of the results (HMM states), we ran the model 4 times (always with 6 states), as suggested by ref. [29]. Additionally, we ran the model with different numbers of states (4, 6, 8, 12). In either case, we visually inspected the states to confirm that the states presented the same spatio-spectral features across runs. We also computed the temporal characteristics of the states over the concatenated recordings, and we compared our results with the TDE-HMM MEG literature.

**Temporal dimension—statistics.** The temporal behavior of the states can be described by computing their temporal properties (lifetime, fractional occupancy, and interval time) or via event-related analysis of the states time course, which provides information about the timing of the bursts and their task-related modulations[26,29]. Considering task data, the information on the sequencing of events is crucial to describe the bursts of activity and link them to different cognitive stages unfolding throughout the task. This last aspect represents also one of the main goals of our study. Therefore, we include in the main manuscript only the event-related analysis, and the temporal properties are reported in the supplementary materials section 3, (Figs. S4 and S5).

Figure 1a reports the event-related field analysis[29]. The time course of each state is epoched with respect to the stimulus onset, taking an epoch of 1400 ms long, [−200 1200] ms. Each trial was baseline corrected considering the pre-stimulus window [−200 −30] ms. Afterward, we ran a two-level generalized linear model (GLM) to investigate the task-dependent changes in the statewise activation pattern. The GLM design matrix consisted of 7 regressors: the constant regressors (average activity overall task conditions), 0 back target, 1 back target, 2 back target, 0 back distractor, 1 back distractor, 2 back distractor—the 6 paradigm conditions. Additional contrast regressors evaluate the effect of response (target vs distractor) and working memory load (0, 1, and 2 back). The GLM first computed the contrast of parameter estimates (COPEs) for each subject (first level), and afterward, the

mean COPEs per subject were fitted across subjects (second level). At the group level, we tested whether the mean COPE at each timepoint significantly differed from zero with a non-parametric permutation test (1e3 permutations). The significance threshold was set to 97.5% of the null distribution, and the multiple comparison correction was carried out via maximum statistics across time and states. For additional details on this methodology, we refer to ref. [29].

## Spectral dimension

*Power spectral density and Coherence.* As described in Fig. 1b, the concatenated original matrix is weighted by the statewise time course—the posterior probabilities. In this way, we obtain a version of the MEG data that describes the activation of each state. Next, we apply a non-parametric multitaper estimation of spectral density, for each state and subject individually, in the broad frequency band 1–40 Hz[24,25]. We obtained the PSD, power distribution over the brain, from which we compute the coherence, resulting in the statewise connectivity matrix. For visualization, the PSD maps are normalized (*z*-score); instead, the coherence matrix is thresholded applying a Gaussian mixture model (GMM) to identify, across subjects, the strongest connections characterizing the statewise phase-coupling network[25,29].

*Spectral decomposition—frequency modes.* This work aims to explore the brain dynamics of working memory in a fully data-driven way, and this also holds when inspecting the states' spectral content. Instead of the conventionally defined frequency rhythms (theta, alpha, beta, etc.), we factorized the spectral content in 4 data-driven frequency modes via NNMF, following[29]. The factorization is run across all subjects, states, nodes (42 parcels), and connections (number of connections = 42 x (42 − 1)/2). By multiplying the PSD and coherence with each frequency mode, we obtain the spectral quantities for each subject and state. The group PSD and coherence are extracted by averaging across subjects.

**Reporting summary**. Further information on research design is available in the Nature Portfolio Reporting Summary linked to this article.

## Data availability

The MEG and MRI data for this study are not publicly available due to privacy restrictions. Researchers interested in collaborating on these data are welcome to contact the senior authors (Prof. Jeroen Van Schependom and Prof. Guy Nagels). The source data to reproduce the figures in this paper are included in the Supplementary Data 1–4.

## Code availability

The analyses were conducted in MATLAB, utilizing the freely accessible HMM-MAR package which can be found here: https://github.com/OHBA-analysis/HMM-MAR. This package belongs to the OSL (OHBA Software Library) toolbox that can be consulted here: https://ohba-analysis.github.io/osl-docs/. In particular, the analysis we implemented was based on the work presented by Quinn et al.[29]. The scripts containing the full pipeline (MEG preprocessing, HMM inference, and data analysis with GLM and spectral decomposition), and that can be used to reproduce the analysis conducted in this work, can be found here: https://github.com/OHBA-analysis/Quinn2018_TaskHMM. For more details on the analysis scripts, contact the corresponding author (Chiara Rossi).

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

## Acknowledgements
The authors would like to thank the participants for their time and commitment to this study. The MEG data collection was enabled by grants from the Belgian Charcot Foundation and by an unrestricted research grant provided by Genzyme-Sanofi. C.R. is funded by Fonds Wetenschappelijk Onderzoek (FWO, Grant numbers: 11K2823N, 11K2821N).

## Author contributions
C.R. conducted the analysis and wrote the manuscript. J.V.S. was the main supervisor of the work and helped write and review the manuscript. D.V., L.C., and G.N. gave inputs for the analysis and provided feedback in the writing process. M.W. and F.A. provided comments on the work. All authors approved the submitted version.

## Competing interests
The authors declare no competing interests.
