## [Peer Review File · Communications Biology]

Reviewers' comments:

Reviewer #1 (Remarks to the Author):

This study Rossi and coll use an data-driven analysis method (TDE-HMM), previously used by co-authors to successfully isolate spontaneous intrinsic networks such as DMN (Vidaurre et al 2018), to identify different brain networks involved during a classical n-back working memory task. These networks are characterized in time, frequency and space. The analysis shows that 4 large networks produce differentiated spectral activations and cortical connectivity patterns during the 500ms time-window post-stimulus presentation. These four networks are coherent with previous findings and suggest that the results are sound, yet not completely novel either. The correspondence with Baddeley's model components is fortunate, and illustrates the usefulness of the analysis method to find functional networks. It would be interesting to use this method in non-classical behavioral tasks/paradigms.

I have only minor comments :

1. Line 67: The statement that "network dynamics unfold over a few milliseconds" seems a bit over-reductive and may give a false idea of what type of network are dealt with , as a) networks are identified over >50ms time intervals and b) intuitively it may take more than just ~5 milliseconds to organize and coordinate large networks spanning various gyri and lobes.
2. Line 461: "Future studies..." sentence is missing a verb.
3. Line 401: Selective alpha power decreases in fusiform cortex are also observed for word awareness in reading (MEG, Levy et al, Cerebral Cortex 2015).
4. Line 415: "the suppressed excitatory beta activity..." reads strangely. How can we know it is excitatory if it appears suppressed? Just suppressed, as vs zero ref baseline.
5. Line 473: typo: Sternberg, with the "r";
6. General comment: The authors mention the "noisy gamma", but it's not clear to which gamma interval they are referring to. Did authors consider analyzing broadband gamma range (for example 50-150Hz) (see Westner, Dalal et al, Plos Comp Biol 2018)?
7. Line 342: typo, Glass brain not brain glass.

Reviewer #2 (Remarks to the Author):

This study undertook a novel analysis of the magnetoencephalographic (MEG) data recorded while participants performed a working memory (n-back) task. A hidden Markov modeling (HMM) approach with time-delay embedding (TDE) has the advantage of detecting brief states in electrophysiological fluctuations when the activity is phase-locked across specific brain regions. The HMM that inferred six states (6 types of repetitive spatiotemporal patterns) was chosen and included four states that are discussed. The primary analysis focused on how the HMM-state occurrence rate varied over time after the stimulus presentation. The authors noted the classic Baddaleley's model of working memory with central executive, phonological loop, and visuospatial sketchpad components. They aimed to map their electrophysiological findings onto this model.

I agree with the authors that the HMM approach can reveal new information about task-evoked neurocognitive processes. Nonetheless, I have a few questions about the event-related analysis approach taken by the authors and their interpretation of the results.

1. I am not sure there is sufficient evidence to relate the reported results to specific neurocognitive processes related to working memory. For example, the authors claim that State 1 can be linked to the central executive. How is this link established? The occurrence of this state was highest 200ms after the letter presentation, but this occupancy was not different between n-back conditions (was not modulated by the demand on the central executive). What's more, the study by Quinn et al. doi: 10.3389/fnins.2018.00603 reports an HMM-TDE state with similar topography/properties and the

highest occupancy at 200ms after viewing faces – figure 5C. An example of evidence for a link between the central executive construct and electrophysiological findings can be found here: <https://doi.org/10.1016/j.ijpsycho.2005.03.018>; manipulation vs. retrieval modulated the EEG.

2. The rate of occurrence of occipital and sensorimotor states (States 2&3) decreased below the baseline during the post-stimulus epoch. Do I understand correctly: When the HMM-TDE states are detected, we can conclude that there is phase-synchrony among the involved regions; but when the occupancy of a given HMM state is below baseline, we do not have sufficient information to interpret this as a suppressed activity? In the latter case, is it not a null result that could correspond to a variety of activity patterns during the time window under consideration? Additional analyses would be needed to understand why the occipital and sensorimotor states that are well represented during resting state are suppressed when visual stimuli are presented and motor response is required (respectively).

3. State 5 is interpreted as the M300 state. This state showed modulation of the post-stimulus occupancy by the working memory load. In my opinion, such modulation is a pre-requisite result to be able to link the HMM states to the working memory processes. Can we interpret this result as showing that the demand on working memory led to an increased occurrence of phase-synchronized activity bursts indexed by the State 5 visits? Accordingly, I would suggest the presentation of this result in the main manuscript (not the supplement).

4. The HMM states are brief (~73ms-long); the MEG + HMM gives us this amazing temporal resolution. Nonetheless, the event-related analysis clumps the time-varying state occurrences into time courses reminiscent of event-related potentials (ERPs) that are of considerably lower resolution (e.g., smooth wave over 400ms). What can we learn about the working memory dynamics if we analyze the brief burst properties, e.g., similar to <https://doi.org/10.1007/s10548-019-00745-5>?

5. It's worth remembering that the source localization of the MEG is inherently ambiguous. It would help to note this to the readers, especially when comparing the advantages of the fMRI and MEG/HMM.

6. A small note: on page 5: MEG was band-filtered 1-45Hz. Why was a notch filter at 50Hz needed?

7. The HMM-TDE was not run with more than 8 states. Previously DOI: 10.1038/s41467-018-05316-z, HMM-TDE with 12 states inferred a visual state with the power lower than the baseline and linked to alpha-band activity (figure2). Given the evidence the authors summarize on page 14 (397-408), it could be hypothesized that this visual state may track the 'visuospatial sketchpad' processing during working memory. I would suggest investigating if an HMM with 12 or 14 states would yield this state of interest in this n-back dataset.

We want to thank the reviewers for the constructive and helpful feedback. We replied to all questions
and extensively elaborated all the answers. We hope the manuscript has become clearer, more
accessible, and more complete.

The most significant change to the manuscript regards the rewriting of the discussion of states 1 and
5. Instead of linking one state to a neuropsychological process of WM, we have interpreted the
different states based on their spatio-spectral features and the WM literature.

We also significantly elaborated the interpretation of the results regarding the event-related analysis
and clarified our choices regarding the analysis of the spectral and temporal dimensions.

Finally, we also performed additional analyses to investigate the states' temporal properties and reran
the model with N=12 HMM states, which we included in the supplementary materials. We also added
the analysis regarding the statewise ER modulation with respect to changing task conditions (target
versus distractor) and increasing working memory load in the main manuscript, section 3.6. Including
this analysis helped interpret and study the different states and their role in WM processing.

In this document, you will find your comments/suggestions (black), as well as our replies (blue) and
the changes to the text (green). We also included the updated figures in this document.

We look forward to receiving your feedback,

Chiara Rossi
On behalf of all authors

Reviewer #1 (Remarks to the Author):

This study Rossi and coll use an data-driven analysis method (TDE-HMM), previously used by co-
authors to successfully isolate spontaneous intrinsic networks such as DMN (Vidaurre et al 2018), to
identify different brain networks involved during a classical n-back working memory task. These
networks are characterized in time, frequency and space. The analysis shows that 4 large networks
produce differentiated spectral activations and cortical connectivity patterns during the 500ms time-
window post-stimulus presentation. These four networks are coherent with previous findings and
suggest that the results are sound, yet not completely novel either. The correspondence with
Baddeley's model components is fortunate, and illustrates the usefulness of the analysis method to
find functional networks. It would be interesting to use this method in non-classical behavioral
tasks/paradigms.

I have only minor comments :

1. Line 67: The statement that "network dynamics unfold over a few milliseconds" seems a bit over-
reductive and may give a false idea of what type of network are dealt with , as a) networks are
identified over >50ms time intervals and b) intuitively it may take more than just ~5 milliseconds to
organize and coordinate large networks spanning various gyri and lobes.

**Response:** Thank you for highlighting this aspect. We agree on the misleading nature of the sentence,
and we modified this and similar ones throughout the whole manuscript.

Instead of referring to the temporal resolution of the MEG data, we chose to bring the focus on the
time window of interest, the epoch. Therefore, we rephrase the aim of our work as: unravelling the
working memory network dynamics in the short time window – an epoch of 1.4 s - in which the WM
processes unfold during an n-back task. As such, we do not lead the reader to expect states with

milliseconds life time, but rather bursts of activity that unfold within an epoch that overall last about
1.4 s.

Example: LINES 80-81

*"In one epoch of 1.4 seconds, we identify data-driven task-relevant states that portray how WM*
*processes unfold."*

2. Line 461: "Future studies..." sentence is missing a verb.

**Response:** this was changed to:

LINE 529-531

*"Future studies should investigate the cross-frequency coupling within each state, in particular, for*
*state 5, in which the broadband activity might hide cross-frequency coupling."*

3. Line 401: Selective alpha power decreases in fusiform cortex are also observed for word
awareness in reading (MEG, Levy et al, Cerebral Cortex 2015).

**Response:** Thank you for pointing out this interesting reading. We have included this reference to
interpretation of state 2 as a phonological loop considering the verbal nature of the stimulus.

LINE: 486-487

*"The alpha activity in the temporal and occipital fusiform regions was observed to decrease with*
*increasing local letter processing and word awareness^{1,2}."*

4. Line 415: "the suppressed excitatory beta activity..." reads strangely. How can we know it is
excitatory if it appears suppressed? Just suppressed, as vs zero ref baseline.

**Response:** Thank you for suggesting this change. We agree with this remark and have changed the
sentence as follows:

Line: 508-509

*"The suppressed beta activity in state 3 prevents the ongoing encoding of WM representation from*
*disruption^{3,4}."*

5. Line 473: typo: Sternberg, with the "r";

**Response:** Changed accordingly.

6. General comment: The authors mention the "noisy gamma", but it's not clear to which gamma
interval they are referring to. Did authors consider analyzing broadband gamma range (for example
50-150Hz) (see Westner, Dalal et al, Plos Comp Biol 2018)?

**Response:** Thank you for addressing this unclear point in the paper. Our study investigates large-scale
brain networks, which have been primarily associated with lower frequency bands (theta and alpha)
98 ^{5,6}. For this reason, in the MEG preprocessing, we focused on the [1-45] Hz frequency range that
includes the low-gamma band [25-45] Hz. This band is associated with the activity captured by spectral
mode 4 in the NNMF (Supplementary Materials Figure S3). We elaborated further this point in the
following paragraph:

LINES: 311-317

*“Spectral mode 4 includes activity within the low-gamma spectrum (25-45 Hz). The model we*
*implemented, the TDE-HMM, is biased towards low frequencies because of the use of PCA, which*
*discards higher frequencies²³. However, intermediate, non-discarded frequencies might be*
*overemphasised with respect to the lowest frequencies³². We then decided to disregard this spectral*
*mode in our interpretation because of this methodological consideration and our focus on larger-scale*
*brain networks. In fact, long-distance connections and information flow in WM have been primarily*
*associated with the synchronizing oscillatory activity of distinct neural populations in lower rhythms*
*(theta and alpha)^{5,6}.”*

Nonetheless, we recognize the important role of gamma in WM as explored in several
neurophysiological studies⁷⁻⁹. Discarding the gamma band in our study entails a methodological
limitation that we report in the limitation section:

LINES: 526-528

*“Gamma band has been consistently reported to play an important role in WM maintenance. Because*
*of methodological limitations we discarded this frequency range. However, future work may*
*investigate the occurrence and role of gamma in the WM network dynamics.”*

7. Line 342: typo, Glass brain not brain glass.

**Response:** Changed accordingly.

Reviewer #2 (Remarks to the Author):

This study undertook a novel analysis of the magnetoencephalographic (MEG) data recorded while
participants performed a working memory (n-back) task. A hidden Markov modeling (HMM)
approach with time-delay embedding (TDE) has the advantage of detecting brief states in
electrophysiological fluctuations when the activity is phase-locked across specific brain regions. The
HMM that inferred six states (6 types of repetitive spatiotemporal patterns) was chosen and
included four states that are discussed. The primary analysis focused on how the HMM-state
occurrence rate varied over time after the stimulus presentation. The authors noted the classic
Baddale’s model of working memory with central executive, phonological loop, and visuospatial
sketchpad components. They aimed to map their electrophysiological findings onto this model.

I agree with the authors that the HMM approach can reveal new information about task-evoked
neurocognitive processes. Nonetheless, I have a few questions about the event-related analysis
approach taken by the authors and their interpretation of the results.

1. I am not sure there is sufficient evidence to relate the reported results to specific neurocognitive
processes related to working memory. For example, the authors claim that State 1 can be linked to
the central executive. How is this link established? The occurrence of this state was highest 200ms
after the letter presentation, but this occupancy was not different between n-back conditions (was
not modulated by the demand on the central executive). What’s more, the study by Quinn et al. doi:
10.3389/fnins.2018.00603 reports an HMM-TDE state with similar topography/properties and the
highest occupancy at 200ms after viewing faces – figure 5C. An example of evidence for a link
between the central executive construct and electrophysiological findings can be found
here: <https://doi.org/10.1016/j.ijpsycho.2005.03.018>; manipulation vs. retrieval modulated the EEG.

**Response:** We thank the reviewer for this constructive comment. We agree that the link between
State 1 and the central executive unit (as well as the interpretation of other states) might seem
speculative. We took this remark as an opportunity to improve our discussion. The answer we provide
covers the following points:

- 1. Change in perspective to discuss the results and clarification on the n-back task.
- 2. the interpretation of the central executive.
- 3. The resemblance of state 1 with a state in Quinn’s work.

**1. Change in perspective to discuss the results and clarification on the n-back task.**

In the suggested paper, Sauseng and colleagues demonstrated that the central executive processes
are executed by frontoparietal theta coherence thanks to the modulation of this mechanism between
two distinguished parts of the task: the manipulation and the retrieval stages¹⁰. In the n-back task, we
cannot make this clear distinction between different stages (encoding, manipulation, and retrieval),
and all the WM processes unfold in a single trial. Therefore, we rewrote the discussion, and instead of
discussing the different WM constructs and linking each to the different states, we now discuss each
state and describe potential interpretations. We clarified this aspect in the following paragraph at the
beginning of the discussion:

Lines: 407-410

*“The n—back task has repeatedly proved to elicit a robust neural activity consistent within and*
*between MEG recording sessions^{11–13}. The task design is such that all the WM processes unroll in a*
*single time window, as opposed to other tasks, such as the Stenberg task, that are designed to probe*
*distinct subprocesses. Therefore, we cannot univocally associate one state with a single WM process.*
*However, we interpret each state starting from its spatio-spectral traits and their functions in the WM*
*literature.”*

As an example of how we changed perspective in discussing the results, we include the revised
discussion on State 1:

LINES 412-439:

*“This state depicts the early (ca 180 ms after stimulus onset) rising of low-frequency activity in the*
*prefrontal, anterior temporal, and posterior cingulate (PCC) regions. We associate the low-frequency*
*mode with the theta (4-8 Hz) rhythm, consistently reported in the WM literature¹⁴⁻¹⁶. Mainly detected*
*in the prefrontal cortex and hippocampus, WM theta activity has been related to integrating and*
*controlling functions during higher-order cognitive processing^{8,9,17}. State 1 exhibits this prefrontal theta*
*activity, suggesting a role as an early high-order stimulus processor and encoder.*

*Interestingly, state 1 displays theta synchronization between the orbitofrontal cortex, OFC, and PCC.*
*This has previously been interpreted as a volitional top-down attentional control mechanism¹⁸. This*
*OFC-PCC connection is also detected in the anterior default mode network (DMN) extracted in resting-*
*state data¹⁹. fMRI WM studies have extensively explored the role of the DMN in WM, reporting this*
*network as actively engaged during WM encoding^{20,21}. Furthermore, the OFC is also theta-coupled with*
*anterior temporal regions, identified as the maintenance regions for verbal stimuli²². This coupling*
*could then embed the attentional control of the OFC onto the temporal cortex during WM maintenance,*
*as previously reported by²³.*

*Therefore, we hypothesize that state 1 extracts the mental representation of the stimulus during*
*encoding and exerts top-down attentional control onto low-level stimulus processing (PCC) and*
*maintenance (temporal cortex). In Baddeley’s neuropsychological model, these functions (among*
*others such as inhibition, manipulation, and shifting) are assigned to the executive control unit^{24,25}, in*
*which Cowan identifies attention as the core component²⁶.*

*We observe that the task-evoked response of state 1 does not significantly modulate with WM load.*
*Instead, a small modulation appears in the contrast between target and distractor trials in the 0 back.*
*This paradigm condition resembles the oddball task, designed to investigate attention. We hypothesize*
*that state 1 might be more driven by attentional requirements, which do not change significantly*
*between different conditions in the n-back paradigm, rather than working memory processing*
*demand, that, instead, we associate with state 5 (as discussed below).*

*The network configuration reveals connections between regions with different roles (for example, OFC*
*attention and anterior temporal maintenance) that could depict the interplay between different*
*processes. This reflects the overlapping unrolling of WM processes during the n-back task and might*
*represent the dynamic communication between the executive unit and the slave components in the*
*neuropsychological model.”*

**2. The interpretation of the central executive.**

In the revised document, we did not link state 1 to the central executive, but rather to specific
functions (encoding and attentional control) that are usually associated with the central executive unit
in Baddeley’s neuropsychological model. However, additional executive functions (manipulation and
updating during retrieval stage) are, instead, carried out by state 5. Therefore, we added this
paragraph to explain the new interpretation of these states:

LINES 441-448

*“State 5 exhibits a broad spectral activity distributed in a complex network involving frontal, temporal,*
*and parietal regions. Like state 1, state 5 also recruits frontal regions, in particular the dorsolateral PFC,*
*with rising theta activity, which are associated with high-order cognitive functions and largely recruited*
*in WM²⁷. Scharinger et al. suggested that the n-back task requires high executive processing demand*

compared to other WM tasks²⁸, which could explain the sustained recruitment of the prefrontal
cortices throughout the epoch in states 1 and 5. However, the functions carried out by the two states
differ depending on the timing of activation, the frequency-specific connections formed between
prefrontal and other regions, and the statewise modulation throughout the task.”

Following, in LINE: 478-479

*“In conclusion, we could theorize that the executive unit of WM could be depicted by two states: state*
*1 for early stimulus encoding and attentional control, and state 5 for manipulation and response*
*processes.”*

**3. The resemblance between state 1 and a state in Quinn’s work**

In the revised manuscript, we have discussed the role of state 1 as an attentional unit performing high-
order stimulus encoding. These functions are required to perform several cognitive tasks, from
working memory to face recognition. Therefore, it is reasonable to expect that a state with the same
spatio-spectral traits as our state 1 would appear analyzing other tasks, and that state would cover
the same role as attentional unit. We hypothesize that the same attentional and stimulus encoding
functions are required both during the n-back task as well as the normal vs scrambled face detection
as described by Quinn et al.²⁹; for this reason, we detect a state with the same spatio-spectral features.

2. The rate of occurrence of occipital and sensorimotor states (States 2&3) decreased below the
baseline during the post-stimulus epoch. Do I understand correctly: When the HMM-TDE states are
detected, we can conclude that there is phase-synchrony among the involved regions; but when the
occupancy of a given HMM state is below baseline, we do not have sufficient information to
interpret this as a suppressed activity? In the latter case, is it not a null result that could correspond
to a variety of activity patterns during the time window under consideration? Additional analyses
would be needed to understand why the occipital and sensorimotor states that are well represented
during resting state are suppressed when visual stimuli are presented and motor response is
required (respectively).

**Response:** As a first step to answering this comment, we have included a new Figure 3a (reported
below) in the main manuscript, which visually describes the steps from the complete statewise time
course to the ER waves. The plot on top shows the probability of activation of all the states throughout
a short time window. Here, we observe that all states inferred by the model are activated and well-
represented throughout the data.

When the occupancy of a particular state goes below baseline, we do have sufficient information to
interpret this as suppressed activity. To clarify this, we added a paragraph to explain better the
meaning of the activation/deactivation of a particular state:

LINES 300-304

*“The increased or decreased activity is observed individually for each state, and it describes the*
*modulation of the statewise occupancy level after the stimulus onset compared to the baseline level.*
*When the task-evoked activity of a state with a strong phase-synchrony goes below baseline, other*
*states (with weaker phase-synchrony in the same frequency band) are activated more frequently.*
*Therefore, the average estimated phase-coupling in the determined frequency range at that time point*
*is below average.”*

For example, state 2 is significantly deactivated around 300 ms and it shows an alpha occipital activity.
In the same time point, state 5 is significantly activated, but state 5 has a weaker alpha activity than
state 2, therefore, the average alpha phase-coupling is reduced – suppressed - in that time point.

This suppressed activity of state 2 is consistent with the traditional event-related desynchronization
 of alpha occipital activity repeatedly detected during WM encoding^{30,31}. As we describe in the
 following paragraph:

 LINE: 482-493
 *“The state’s evoked-response is significantly decreased between 200 and 500 ms PST (Figure 4b),*
 *resembling the event-related desynchronisation (ERD) of the occipital alpha activity that several WM*
 *neuroimaging studies detected during and following the early stimulus encoding phase^{28,31–33}. The*
 *alpha activity in the temporal and occipital fusiform regions was observed to decrease with increasing*
 *local letter processing and word awareness^{1,2}.*

...
 *Therefore, the suppressed alpha in state 2 could reflect local independent letter processing, as*
 *traditionally reported in the WM literature. The evoked-response of state 2 is modulated by WM load;*
 *with increasing WM load - increasing processing demand -, the state 2 is suppressed for a longer time.*
 *The same effect is consistently observed for the alpha occipital ERD wave³¹, and this observation*
 *corroborates the interpretations of state 2 as local letter processing.”*

 Regarding the activity of the sensorimotor network, we observe that the state 3 is suppressed during
 stimulus encoding. However, the SMC is also recruited in state 5, which we discussed being involved
 in the response selection process:

 LINES: 472-476
 *The activation of state 5 is significantly amplified and prolonged in target as compared to distractor*
 *trials (Figure 6a), and this difference might reflect the passage from perceptual to motor processing*
 *after target recognition. In fact, we observe that parietal regions (such as the medial SMC and the*
 *supramarginal regions) displaying beta activity are also recruited in state 5, and they have previously*
 *been detected in response selection and motor planning³.*

**Figure 3 Pipelines to extract the states temporal, spectral, and spatial profiles. a) Temporal dimension.** Starting from the states time courses
– the posterior probabilities -, we cut the time series with respect to the task information (stimulus onset) to define the trials, time windows
[-02, 1.2] s. We perform baseline correction. Next, we ran the generalized linear model (GLM) to statistically evaluate the increased or
decreased modulation of the statewise fractional occupancy (activation) level compared to baseline. We report the states' average response
across conditions (provided by the constant regressor). The straight lines at the bottom indicate the time points where the state of the same
color is significantly activated or deactivated (permutation test, significance level 0.025). The significantly modulated, task-relevant states
are state 1, state 2, state 3, and state 5. **b) Spectral dimension.** We weighted the MEG recordings by the states time courses. Afterward, we
used a multitaper to compute the spectral density of the weighted MEG data for each subject and state separately. We then extracted the
power spectral density (PSD) over the brain, which constitutes the spatial map of activation of a state, and the coherence across regions
which constitutes the phase-coupling network of a state. We reported the plot of the statewise PSD averaged across regions over the broad
frequency spectrum (1-40 Hz) – the bold lines display the mean across subjects, and the lighter area includes the standard deviation across
subjects. The same plot is reproduced for the phase-coupling averaged over all the connections in the broadband spectrum (1-40 Hz) – the
bold lines show the mean phase-coherence across subjects, and the area includes the standard deviation of the group.

3. State 5 is interpreted as the M300 state. This state showed modulation of the post-stimulus
occupancy by the working memory load. In my opinion, such modulation is a pre-requisite result to
be able to link the HMM states to the working memory processes. Can we interpret this result as
showing that the demand on working memory led to an increased occurrence of phase-synchronized
activity bursts indexed by the State 5 visits? Accordingly, I would suggest the presentation of this
result in the main manuscript (not the supplement).

**Response:** Thank you for raising this point. We recognize the importance of the analyses showing the
WM modulated activity of the different states, and we included the results in the main manuscript
(manuscript Section 3.6, Figure 6):

**LINES: 376-391**

***“Target vs Distractor trials***

*Figure 6a shows the difference between the states' task-evoked response during target and distractor*
*trials. State 1, the prefrontal theta state, presents a small but significantly higher peak of activation at*
*200 ms in the 0 back target compared to the 0 back distractor condition. State 3, the sensorimotor*
*state, shows a significantly decreased task-evoked response in the target compared to distractor trials*
*around 400 ms in the 0 and 1 back conditions, and between 400 and 700 ms in the 2-back condition.*
*Last, state 5, the M300 state, presents a significantly amplified (+20 %) task-evoked response in target*
*than distractor trials between 300 ms and 700 ms in all WM load conditions.*

***WM load***

*Next, we investigated the difference between the states' task-evoked response between WM load*
*conditions (1-0, 2-0, and 2-1), Figure 6b. The analysis incorporated all the target and distractor trials*
*for each WM load condition. The activation of state 5 is significantly increased in the 1 back condition*
*compared to 0 back condition around 300 ms, also in the 2 back condition compared to both 1 and 0*
*back conditions between 400 and 1000 ms after stimulus onset. Instead, the occupancy level of state 2*
*is significantly reduced in the 2 back condition compared to 0 and 1 back conditions between 650 ms*
*and 750 ms after stimulus onset.*

MAIN MANUSCRIPT Figure 6 States' task-evoked activity modulated between paradigm conditions. **a) Target versus Distractors.** The graphs report the contrast regressors to evaluate the difference in states' activation between target (T) and distractor (D) trials. From left to right, each plot considers a single load condition: 0, 1, and 2. **b) WM load.** The plots show the contrast regressors evaluating the difference between all pairs of WM load conditions: 0, 1, and 2. The straight lines at the bottom of each graph sign the time points when the state's contrast regressor (of the same color) is significantly modulated compared to baseline.

We have re-elaborated the discussion on state 5 following this comment and the first remarks made by the reviewer. For the approach used to write the discussion, we refer to the answer we provided to the first remark.

We added the following paragraph in the main manuscript to discuss the task-based modulated activity of state 5:

LINES: 458-465

“The task-locked occupancy level of state 5, the M300, is modulated by WM load (Figure 6b). The increasing task difficulty requires increasing resources in the matching and recalling processes that we observe resulting in an increased M300 amplitude. Jensen et al. have reported an increased theta activity with increasing WM load during the retention stage¹⁵, not in the early encoding step, supporting the increased M300 amplitude and justifying the different behavior we observe between state 1 and 5. While the M300 wave shows an increased amplitude, the P300 amplitude decreases with increasing WM load^{28,34,35}, and this opposite effect might lay in the in the physical nature of the signal, or in the different perspective of analysis (source-reconstructed MEG functional networks versus sensor-level single region EEG data).”

4. The HMM states are brief (~73ms-long); the MEG + HMM gives us this amazing temporal resolution. Nonetheless, the event-related analysis clumps the time-varying state occurrences into time courses reminiscent of event-related potentials (ERPs) that are of considerably lower resolution (e.g., smooth wave over 400ms). What can we learn about the working memory dynamics if we analyze the brief burst properties, e.g., similar to <https://doi.org/10.1007/s10548-019-00745-5?>

Response: We thank the reviewer for this comment. First, we added the following paragraph (methods section) to clarify how the temporal dimension can be investigated and justify our choice to

392 include only the ER analysis in the main manuscript and report the temporal properties in the
393 supplementary materials:

LINES: 222-228

*“The temporal behavior of the states can be described by computing their temporal properties (life*
*time, fractional occupancy, and interval time) or via event-related analysis of the states time course,*
*which provides information over the timing of the bursts and their task-related modulations^{29,36}.*
*Considering task data, the information on the sequencing of events is crucial to describe the bursts of*
*activity and link them to different cognitive stages unfolding throughout the task. This last aspect*
*represents also one of the main goals of our study. Therefore, we include in the main manuscript only*
*the event-related analysis, and the temporal properties are reported in the supplementary materials*
*section 3, Figures S4-S5.”*

Temporal Properties

The temporal properties of the bursts can be evaluated over the concatenated data or at the trial
level. In the first version of the paper, we only computed the temporal properties of the states over
the concatenated data, discarding the task information (supplementary Figure S4). The goal of this
analysis was to verify that the temporal properties were consistent with what previous HMM studies
had reported.

We include the following paragraph in the revised manuscript, LINES: 288-294

*“To test the model reliability, we visually assessed the results of 4 different inferences (with 6 states),*
*as suggested by²⁹, and concluded that the model could consistently infer states with similar spatio-*
*spectral traits. Additionally, we computed the temporal characteristics (life time, LT, interval time, IT,*
*and fractional occupancy, FO) of the states over the concatenated data to verify that these results are*
*consistent with what was previously reported by different HMM studies. We report this analysis in the*
*supplementary materials (section 3, Figure S4). The states’ average life time is 73 ms, the average*
*fractional occupancy is 18%, and the average interval time is 500 ms. These results are consistent with*
*the HMM literature on resting-state and task data^{29,36,37}.”*

We have then evaluated the bursts' temporal properties at the trial level. We computed the life time
(LT) and the fractional occupancy (FO) of the states. We didn't evaluate the interval time because,
from the analysis over the concatenated dataset, we observe that the statewise IT is 500 ms,
therefore, we expect each state to appear once per trial.

We elaborated the following paragraph in the Supplementary Section 3 Figure S5:

LINES: 76-86 supplementary materials:

*“Following, we computed the LT and IT per paradigm condition. Starting from the Viterbi path, each*
*parameter was extracted per state, trial, and subject, and then averaged across trials per subject and*
*paradigm condition. Supplementary Figure S5 reports the statewise distribution of each parameter*
*over subjects per paradigm condition. The LT and the FO of state 5 become longer with increasing WM*
*load and in target as compared to distractor trials. These results correspond to the results from the*
*evoked-response analysis of the states’ time courses. The latter revealed for state 5 an increased*
*occupancy level with increasing WM load and in target as compared to distractor trials (Figure 6 main*
*manuscript). Instead, the event-related responses of states 2 and 3 display significant task-related*
*modulations which do not find a corresponding significant variation in temporal properties. This*
*suggests that, for states 2 and 3, the sustained changes in evoked response results from the average*
*across trials, as also demonstrated by Quinn³⁶.*
*This analysis represents a first step towards a trial-based analysis, which falls beyond the scope of this*
*manuscript.”*

**Supplementary Figure S4. Temporal properties of the task-relevant states.** On the left the lifetime (LT), and on the right the
 fractional occupancy (FO). The temporal characteristics are computed for each paradigm condition, separately to assess their
 modulation with respect to WM load and target versus distractor. The comparisons are carried out via non-parametric
 Wilcoxon’s rank-sum test, * $p_{value} < 0.05$. All results are corrected for multiple comparison via FDR correction³⁸.

 5. It’s worth remembering that the source localization of the MEG is inherently ambiguous. It would
 help to note this to the readers, especially when comparing the advantages of the fMRI and
 MEG/HMM.

 **Response:** We thank the reviewer for pointing out this aspect. We included this remark in the
 introduction as follows:

 LINES: 74-76

*“In fMRI data, the HMM states resembled the canonical resting-state networks¹⁹. In MEG data,
 although considering the inherent ambiguity of source-reconstructed data, these states can track the
 evolution of cognitive processes with great temporal resolution^{36,39}.”*

 6. A small note: on page 5: MEG was band-filtered 1-45 Hz. Why was a notch filter at 50Hz needed?

**Response:** Thank you for pointing out this unclear step in the preprocessing. The superposition of the
 two filters follows a conservative approach we decided to maintain throughout the data processing.

Figure 1. The two plots present the power spectral density (PSD) for two subjects and two channels, separately. In each graph, the blue line represents the PSD of the signal only after the 1-45 Hz bandpass filter, whereas the orange line shows PSD after the 1-45 Hz bandpass filter and the notch filter at 50 Hz.

Figure 1 reports the frequency spectrum for two subjects (1 and 10) and two channels (300 and 50),
 separately. After applying a (1-45 Hz) bandpass filter, we observed that the contribution of the
 powerline (50 Hz) was still relevant and appears across subjects and channels. We run the model on
 the concatenated recordings of all subjects, and the inferred states arise from reoccurring spectral
 patterns across recordings and subjects⁴⁰. Therefore, we decided to include a notch filter at 50 Hz to
 avoid that one state could pick up this 50 Hz component, or that this contribution could represent a
 source noise in the statewise spectral content.

**LINE: 152-153**

*“We also included a notch filter at 50 Hz to remove the remaining power line effect, which could*
 *represent a source of noise for the HMM inference.”*

7. The HMM-TDE was not run with more than 8 states. Previously DOI: 10.1038/s41467-018-05316-
 z, HMM-TDE with 12 states inferred a visual state with the power lower than the baseline and linked
 to alpha-band activity (figure2). Given the evidence the authors summarize on page 14 (397-408), it
 could be hypothesized that this visual state may track the ‘visuospatial sketchpad’ processing during
 working memory. I would suggest investigating if an HMM with 12 or 14 states would yield this state
 of interest in this n-back dataset.

**Response:** Thank you for raising this point which helped us clarify the role of state 2 (the occipital
 alpha state) and justify the choice of the number of states to infer.

In the revised manuscript, we added the following paragraph to elaborate on the interpretation of the
 occipital alpha state as phonological loop rather than visuospatial sketchpad during a visuo-verbal n-
 back task:

**LINE: 495-501**

*“In the visual verbal n-back task, the verbal nature of the stimulus (the letters) determines how the*
 *information is stored and maintained^{25,41}. As suggested in Baddeley’s neuropsychological model, the*
 *visually acquired letter is translated into its phonological representation and is stored as such in the*
 *phonological loop²⁵. This compartment has been linked to language processing areas (e.g., Broca’s area*
 *and temporal cortex), which are also recruited in state 2⁴². Therefore, in state 2, the alpha phase-*
 *coupling between occipital and temporal regions could represent the first visual (occipital) processing*
 *being translated in – and then stored as - the correspondent phonological representation (temporal).”*

Following this remark, we also ran an HMM inference with 12 states. We reported the results in the
 supplementary materials (section 5, Figure S12). This analysis helps demonstrate the replicability of

the main WM networks that we report in the study, and exemplifies the issues encountered when
increasing the number of states.

In the main manuscript:

LINES:284-288

*“We identified 6 as the optimal number of states as this setting could pick up the expected spatio-*
*spectral traits of the WM task and minimize the redundant information across states³⁷. We report the*
*results of the 12 states inference in the supplementary materials section S5 (Figure S12), to*
*demonstrate the replicability of the relevant states identified in the 6 states inference, and to show the*
*issues encountered with increasing number of states.”*

In the supplementary materials, we elaborated the following paragraphs:

LINES: 144-170

*“Figure S12 reports the results for the 12 states inference. From the event-related (ER) analysis of the*
*states’ posterior probabilities of activation, we identify 6 states with an evoked response that are*
*significantly modulated throughout the epoch. The model extracted 2 occipital states (state 2 and state*
*5) with identical spatial maps and ER profiles. The average power spectral density (PSD) plots related*
*to the two states present a 10 Hz peak, which we then associate with alpha activity. These states*
*correspond to state 2 for the 6 states inference presented in the paper.*

*State 10 embeds the theta prefrontal state that peaks around 200 ms and resembles the theta*
*prefrontal state that we observe in the 6 states inference.*

*States 3 and 8 present an M300 temporal profile. While state 3 shows a broad frontoparietal activation,*
*state 8 seems to recruit more specifically the sensorimotor regions. However, the temporal*
*characteristics and the spectral content of these states are indistinguishable.*

*Lastly, state 11 is significantly suppressed early after stimulus onset and it shows upper frontal beta*
*activity. This beta suppression is what we also observe in the 6 states inference in state 3, which,*
*instead, showed mostly beta activity in the sensorimotor cortex.*

*The task-relevant states detected in the 12 states inference closely resemble the task-relevant states*
*that we reported in our paper, which makes our networks replicable and demonstrates the model*
*reliability. Additionally, we observe two interesting aspects. First, the occurrence of 2 or more states*
*sharing several spatio-spectral features (such as the two occipital states) is a drawback that we risk*
*encountering when increasing the number of states to infer. Secondly, we observe an interesting*
*phenomenon for which our M300 state now seems to split across two states, one specific to*
*prefrontal/temporal functions (resembling the M300 state in the 6 states inference) and one specific*
*to the sensorimotor network. This split might mirror the distinction of two functions (perceptual and*
*motor processing). However, the analysis of these two states is non-trivial because they seem to share*
*temporal and spectral properties.*

*Lastly, this analysis shows a limitation of this method related to the stochasticity of the inference. By*
*running the model several times, the results are not 100% replicable, and some states might differ in*
*spatio-spectral traits, such as state 11 in the 12 states inference (superior frontal beta) and state 3 in*
*the 6 states inference (sensorimotor beta).“*

Supplementary Figure S10. Main results from the 12 states inference. The top left figure reports the average response activation for all 12 states in one epoch (constant regressor of the GLM analysis). Here, we observe 5 states with remarked modulation of their evoked response after stimulus presentation. On the right, the two graphs report the mean power spectral density and phase-coherence for each state, averaged across subjects and regions/connections (second and third plot, respectively). Below, we report the spatial distribution of the z-score power spectral density over the brain, for the relevant states.

**References**

- 1. Costers, L. *et al.* Spatiotemporal and spectral dynamics of multi-item working memory as
revealed by the n-back task using MEG. *Hum. Brain Mapp.* **41**, 2431–2446 (2020).
- 2. Levy, J., Vidal, J. R., Fries, P., Démonet, J. F. & Goldstein, A. Selective neural synchrony
suppression as a forward gatekeeper to piecemeal conscious perception. *Cereb. Cortex* **26**,
3010–3022 (2016).
- 3. Schneider, D., Barth, A. & Wascher, E. On the contribution of motor planning to the retroactive
cuing benefit in working memory: Evidence by mu and beta oscillatory activity in the EEG.
*Neuroimage* **162**, 73–85 (2017).
- 4. Schneider, D., Barth, A., Getzmann, S. & Wascher, E. On the neural mechanisms underlying the
protective function of retroactive cuing against perceptual interference: Evidence by event-
related potentials of the EEG. *Biol. Psychol.* **124**, 47–56 (2017).
- 5. Fries, P. A mechanism for cognitive dynamics: Neuronal communication through neuronal
coherence. *Trends Cogn. Sci.* **9**, 474–480 (2005).
- 6. Wang, R. *et al.* Consistency and dynamical changes of directional information flow in different
brain states: A comparison of working memory and resting-state using EEG. *Neuroimage* **203**,
116188 (2019).
- 7. Heinrichs-Graham, E. & Wilson, T. W. Spatiotemporal oscillatory dynamics during the encoding
and maintenance phases of a visual working memory task. *Cortex* **August**, 121 (2015).
- 8. Palva, S., Kulashekhar, S., Hämäläinen, M. & Palva, J. M. Localization of cortical phase and
amplitude dynamics during visual working memory encoding and retention. *J. Neurosci.* **31**,
5013–5025 (2011).
- 9. Brookes, M. J. *et al.* Changes in brain network activity during working memory tasks: A
magnetoencephalography study. *Neuroimage* **55**, 1804–1815 (2011).
- 10. Sauseng, P., Klimesch, W., Schabus, M. & Doppelmayr, M. Fronto-parietal EEG coherence in
theta and upper alpha reflect central executive functions of working memory. *Int. J.*
*Psychophysiol.* **57**, 97–103 (2005).
- 11. Ahonen, L., Huotilainen, M. & Brattico, E. Within- and between-session replicability of cognitive
brain processes: An MEG study with an N-back task. *Physiol. Behav.* **158**, 43–53 (2016).
- 12. Kane, M. J. & Conway, A. R. A. The invention of n-back: An extremely brief history. *The*
*Winnower* **3**, 2003–2005 (2016).
- 13. Rac-lubashevsky, R. & Kessler, Y. Decomposing the n-back task: An individual differences study
using the reference-back paradigm. *Neuropsychologia* (2016)
doi:10.1016/j.neuropsychologia.2016.07.013.
- 14. Bahmani, Z. *et al.* Prefrontal Contributions to Attention and Working Memory. *Curr Top Behav*
*Neurosci* 129–153 (2019) doi:10.1007/7854.
- 15. Jensen, O. & Tesche, C. D. Frontal theta activity in human increases with memory load in a
working memory task. *Eur. J. Neurosci.* **15(8)**, 1395–9 (2002).
- 16. Scheeringa, R. *et al.* Trial-by-trial coupling between EEG and BOLD identifies networks related
to alpha and theta EEG power increases during working memory maintenance. *Neuroimage*
**44**, 1224–1238 (2009).
- 17. Nuñez, A. & Buño, W. The Theta Rhythm of the Hippocampus: From Neuronal and Circuit
Mechanisms to Behavior. *Front. Cell. Neurosci.* **15**, 1–16 (2021).
- 18. Buschman, T. J. & Miller, E. K. Top-down versus bottom-up control of attention in the prefrontal
and posterior parietal cortices. *Science (80-.)*. **315**, 1860–1864 (2007).
- 19. Vidaurre, D. *et al.* Spontaneous cortical activity transiently organises into frequency specific
phase-coupling networks. *Nat. Commun.* **9**, (2018).
- 20. Linden, D. E. J. The working memory networks of the human brain. *Neuroscientist* **13**, 257–267
(2007).
- 21. Piccoli, T. *et al.* The default mode network and the working memory network are not anti-

- correlated during all phases of a working memory task. *PLoS One* **10**, 1–16 (2015).
- 22. Dimakopoulos, V., Mégevand, P., Stieglitz, L. H., Imbach, L. & Sarnthein, J. Information flows
from hippocampus to auditory cortex during replay of verbal working memory items. *Elife* **11**,
1–19 (2022).
- 23. Daume, J., Graetz, S., Gruber, T., Engel, A. K. & Frieze, U. Cognitive control during audiovisual
working memory engages frontotemporal theta-band interactions. *Sci. Rep.* **7**, 1–13 (2017).
- 24. Baddeley, A. Working memory: Theories, models, and controversies. *Annu. Rev. Psychol.* **63**,
1–29 (2012).
- 25. Baddeley, A. Working memory: Looking back and looking forward. *Nat. Rev. Neurosci.* **4**, 829–
839 (2003).
- 26. Cowan, N. Working memory development: A 50-year assessment of research and underlying
theories. *Cognition* **224**, 105075 (2022).
- 27. Cristofori, I., Cohen-Zimerman, S. & Grafman, J. *Executive functions. Handbook of Clinical*
*Neurology* vol. 163 (Elsevier B.V., 2019).
- 28. Scharinger, C., Soutschek, A., Schubert, T. & Gerjets, P. Comparison of the working memory
load in N-back and working memory span tasks by means of EEG frequency band power and
P300 amplitude. *Front. Hum. Neurosci.* **11**, 1–19 (2017).
- 29. Quinn, A. J. *et al.* Task-evoked dynamic network analysis through Hidden Markov Modeling.
*Front. Neurosci.* **12**, 1–17 (2018).
- 30. Heinrichs-Graham, E. & Wilson W., T. Spatiotemporal oscillatory dynamics during the encoding
and maintenance phases of a visual working memory task. *Cortex* **69**, 121–130 (2015).
- 31. Wianda, E. & Ross, B. The roles of alpha oscillation in working memory retention. *Brain Behav.*
**9**, 1–21 (2019).
- 32. Pfurtscheller, G. Functional brain imaging based on ERD/ERS. *Vision Res.* **41**, 1257–1260 (2001).
- 33. Syrjälä, J., Basti, A., Guidotti, R., Marzetti, L. & Pizzella, V. Decoding working memory task
condition using magnetoencephalography source level long-range phase coupling patterns. *J.*
*Neural Eng.* **18**, (2021).
- 34. Scharinger, C., Soutschek, A., Schubert, T. & Gerjets, P. When flanker meets the n-back: What
EEG and pupil dilation data reveal about the interplay between the two central-executive
working memory functions inhibition and updating. *Psychophysiology* **52**, 1293–1304 (2015).
- 35. Kok, A. On the utility of P3 amplitude as a measure of the processing capacity.
*Psychophysiology* **38**, 557–577 (2001).
- 36. Quinn, A. J. *et al.* Unpacking Transient Event Dynamics in Electrophysiological Power Spectra.
*Brain Topogr.* **32**, 1020–1034 (2019).
- 37. Baker, A. P. *et al.* Fast transient networks in spontaneous human brain activity. *Elife* **2014**, 1–
18 (2014).
- 38. Benjamini, Y. & Hochberg, Y. Controlling the False Discovery Rate: A Practical and Powerful
Approach to Multiple Testing. *J. R. Stat. Soc. Ser. B* **57**, 289–300 (1995).
- 39. Vidaurre, D., Smith, S. M. & Woolrich, M. W. Brain network dynamics are hierarchically
organized in time. *Proc. Natl. Acad. Sci. U. S. A.* **114**, 12827–12832 (2017).
- 40. Vidaurre, D. *et al.* Spectrally resolved fast transient brain states in electrophysiological data.
*Neuroimage* **126**, 81–95 (2016).
- 41. Repovš, G. & Baddeley, A. The multi-component model of working memory: Explorations in
experimental cognitive psychology. *Neuroscience* **139**, 5–21 (2006).
- 42. Chai, W. J., Abd Hamid, A. I. & Abdullah, J. M. Working memory from the psychological and
neurosciences perspectives: A review. *Front. Psychol.* **9**, 1–16 (2018).

REVIEWERS' COMMENTS:

Reviewer #2 (Remarks to the Author):

I would like to thank the authors for their careful consideration of my comments. I greatly enjoyed reading their rebuttal letter. I believe that the manuscript is appropriate for publication.